# A transformation from temporal to ensemble coding in a model of piriform cortex

Merav Stern[1,2†], Kevin A Bolding[3], LF Abbott[2], Kevin M Franks[3]*

[1]Edmond and Lily Safra Center for Brain Sciences, Hebrew University, Jerusalem, Israel; [2]Department of Neuroscience, Zuckerman Mind Brain Behavior Institute, Columbia University, New York, United States; [3]Department of Neurobiology, Duke University School of Medicine, Durham, United States

**Abstract** Different coding strategies are used to represent odor information at various stages of the mammalian olfactory system. A temporal latency code represents odor identity in olfactory bulb (OB), but this temporal information is discarded in piriform cortex (PCx) where odor identity is instead encoded through ensemble membership. We developed a spiking PCx network model to understand how this transformation is implemented. In the model, the impact of OB inputs activated earliest after inhalation is amplified within PCx by diffuse recurrent collateral excitation, which then recruits strong, sustained feedback inhibition that suppresses the impact of later-responding glomeruli. We model increasing odor concentrations by decreasing glomerulus onset latencies while preserving their activation sequences. This produces a multiplexed cortical odor code in which activated ensembles are robust to concentration changes while concentration information is encoded through population synchrony. Our model demonstrates how PCx circuitry can implement multiplexed ensemble-identity/temporal-concentration odor coding.

DOI: https://doi.org/10.7554/eLife.34831.001

*For correspondence:
franks@neuro.duke.edu

Present address: †The Raymond and Beverly Sackler Scholars Program in Integrative Biophysics, University of Washington, Seattle, United States

Competing interests: The authors declare that no competing interests exist.

## Introduction

Although spike timing information is often used to encode features of a stimulus (*Panzeri et al., 2001*; *Thorpe et al., 2001*; *Gollisch and Meister, 2008*; *Zohar et al., 2011*; *Gütig et al., 2013*; *Zohar and Shamir, 2016*), it is not clear how this information is decoded by downstream areas (*Buzsáki, 2010*; *Panzeri et al., 2014*; *Zohar and Shamir, 2016*). In olfaction, a latency code is thought to be used in olfactory bulb (OB) to represent odor identity (*Bathellier et al., 2008*; *Cury and Uchida, 2010*; *Shusterman et al., 2011*; *Gschwend et al., 2012*). This information is transformed into a spatially distributed ensemble in primary olfactory (piriform) cortex (PCx) (*Uchida et al., 2014*). PCx is a three-layered cortex with well-characterized circuitry (*Bekkers and Suzuki, 2013*), providing an advantageous system to mechanistically dissect this transformation. Here, we develop a spiking network bulb-cortex model to examine how temporally structured odor information in OB is transformed in PCx.

In mammals, odor perception begins when inhaled volatile molecules bind to odorant receptors on olfactory sensory neurons (OSNs) in the nasal epithelium. Each OSN expresses just one of ~1000 different odorant receptor genes (*Buck and Axel, 1991*). Odorant receptors are broadly tuned so that OSN firing rates reflect their receptor's affinity for a given odorant and the odorant concentration (*Malnic et al., 1999*; *Jiang et al., 2015*). All OSNs expressing a given receptor converge on a unique pair of OB glomeruli (*Mombaerts et al., 1996*), where they make excitatory synaptic connections onto dendrites of mitral/tufted cells (MTCs), the sole output neurons of the OB. Because each MTC only receives excitatory input from one glomerulus, each MTC essentially encodes the

activation of a single class of odorant receptor. MTCs exhibit subthreshold, respiration-coupled membrane potential oscillations (*Cang and Isaacson, 2003*; *Margrie and Schaefer, 2003*) that may help transform rate-coded OSN input into a temporal latency code in the OB (*Hopfield, 1995*; *Schaefer et al., 2006*; *Schaefer and Margrie, 2012*). Individual MTC responses exhibit odor-specific latencies that tile the ~300–500 ms respiration (sniff) cycle (*Bathellier et al., 2008*; *Cury and Uchida, 2010*; *Shusterman et al., 2011*; *Gschwend et al., 2012*), and decoding analyses indicate that spike time information is required to accurately represent odor identity in the OB (*Cury and Uchida, 2010*; *Junek et al., 2010*). Thus, the OB uses a temporal code to represent odor identity. Olfactory information is conveyed to PCx via MTC projections that are diffuse and overlapping (*Ghosh et al., 2011*; *Miyamichi et al., 2011*; *Sosulski et al., 2011*), ensuring that individual PCx principal neurons receive inputs from different combinations of co-activated glomeruli (*Franks and Isaacson, 2006*; *Suzuki and Bekkers, 2006*; *Apicella et al., 2010*; *Davison and Ehlers, 2011*). Consequently, odors activate distinct ensembles of neurons distributed across PCx (*Illig and Haberly, 2003*; *Rennaker et al., 2007*; *Stettler and Axel, 2009*; *Roland et al., 2017*). Recent studies indicate that odor identity in PCx is encoded simply by the specific ensembles of cells activated during the sniff, with no additional information provided by spike timing (*Miura et al., 2012*; *Bolding and Franks, 2017*). Thus, a temporal odor code in OB is transformed into an ensemble code in PCx. However, these ensembles are sensitive to the sequence in which glomeruli are activated (*Haddad et al., 2013*), indicating that PCx could parse temporally structured OB input. Whether, or how it does so is not known.

Although MTCs respond throughout the respiration cycle, PCx recordings in awake animals indicate that most odor-activated cells respond transiently, shortly after inhalation (*Miura et al., 2012*; *Bolding and Franks, 2017*). Taken together, these data suggest that cortical odor responses are preferentially defined by the earliest-active glomeruli and that glomeruli activated later in the sniff are relatively ineffective at driving responses. To examine this directly, we obtained simultaneous recordings of odor-evoked spiking in populations of presumed MTCs and PCx principal cells in awake, head-fixed mice (*Figure 1*). Consistent with previous studies, a given odor activated a large subset of MTCs, with individual cells responding with onset latencies distributed across the respiration cycle (*Figure 1C*). By contrast, activity was much sparser in PCx, with most responsive cells spiking within 50 ms of inhalation (*Figure 1D*). At the population level, odors evoked a sustained increase in MTC spiking throughout the sniff (*Figure 1E*), while spiking activity in PCx peaks briefly after inhalation followed by a period of sustained suppression (*Figure 1F*). Together with the data discussed above, these results indicate that a spatio-temporal code for odor identity in OB is transformed into an ensemble code in PCx in which the cortical ensemble is largely defined by the earliest-active OB inputs and information conveyed by later-responding OB inputs is discounted.

How is this transformation implemented? Because total OB output is sustained and can even grow over time, suppression of later responses must originate from inhibition within PCx itself. Multiple circuit motifs within PCx are poised to dramatically reshape odor representations. First, MTCs make excitatory connections onto layer one inhibitory interneurons that provide feedforward inhibition (FFI) to pyramidal cells (*Luna and Schoppa, 2008*; *Stokes and Isaacson, 2010*; *Suzuki and Bekkers, 2012*). Pyramidal cells also form a widespread recurrent collateral excitatory plexus that, in turn, recruits strong feedback inhibition (FBI) from a distinct class of layer 2/3 interneurons (*Stokes and Isaacson, 2010*; *Franks et al., 2011*; *Suzuki and Bekkers, 2012*; *Large et al., 2016*), and this intracortical recurrent circuitry is thought to contribute substantially to odor-evoked cortical responses (*Davison and Ehlers, 2011*; *Poo and Isaacson, 2011*; *Haddad et al., 2013*). The specific roles that each of these circuit elements play in shaping cortical odor ensembles is not known.

## Results

To understand how OB input is integrated and transformed in PCx, we simulated patterns of odor-evoked MTC activity over a single respiration cycle and used this as input to a PCx network consisting of leaky integrate-and-fire neurons. We first describe the implementation of the full model and demonstrate that it grossly recapitulates experimental findings. We then examine the specific roles that different circuit components play in generating these responses by exploring how the model behaviors change as the model parameters are varied. We find that PCx odor responses are largely defined by the earliest-active OB inputs, that the impact of these inputs is amplified by recurrent

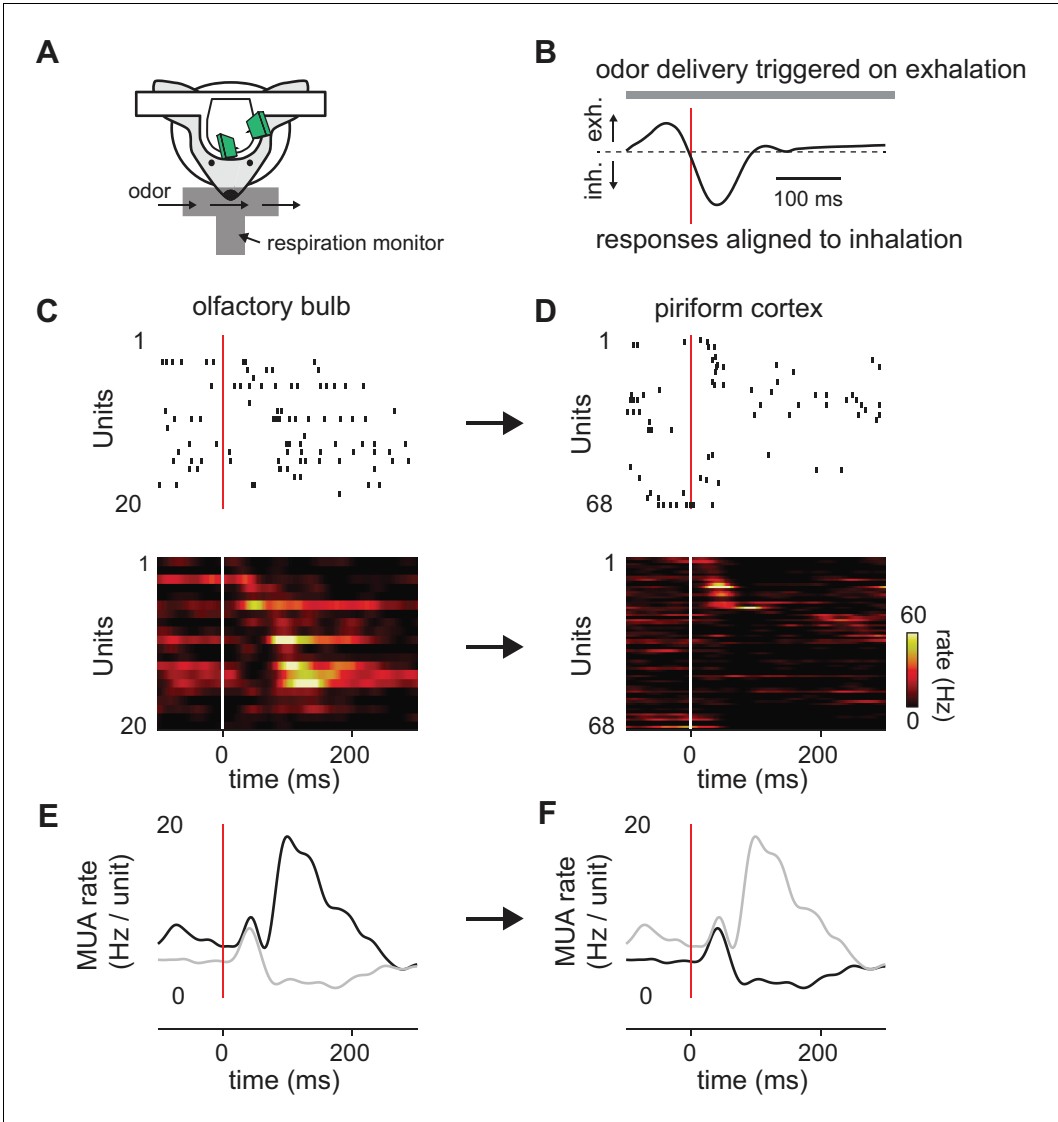

**Figure 1.** Transformation of odor information from OB to PCx. (**A**) Experimental setup. (**B**) Example respiration trace. Odor deliveries (1 s pulses) were triggered by exhalation and trials are aligned to the onset of the next inhalation (red line). (**C,D**) Single-trial raster plots (top) and average firing rates (15 trials, bottom) for simultaneously recorded populations of cells in OB (**C**) and PCx (**D**), during a respiration cycle as in B. Cells are sorted by mean latency to first spike. (**E,F**) Population peristimulus time histograms for the cells shown above (dark traces) in OB (**E**) and PCx (**F**) (dark traces). For comparison, the PSTHs from the other area are overlaid (light traces).
DOI: https://doi.org/10.7554/eLife.34831.002

excitation, while the impact of OB inputs that respond later is suppressed by feedback inhibition. We further find that this configuration supports odor recognition across odorant concentrations, while preserving a representation of odor concentration in the synchrony of the population response.

## Odors activate distinct ensembles of piriform neurons
We simulated OB and PCx spiking activity over the course of a single respiration cycle consisting of a 100 ms exhalation followed by a 200 ms inhalation. Our model OB consisted of 900 glomeruli that are each innervated by a unique family of 25 mitral cells. Odor identities are defined by sets of glomerular onset latencies because different odors activate specific subsets of glomeruli with odor-

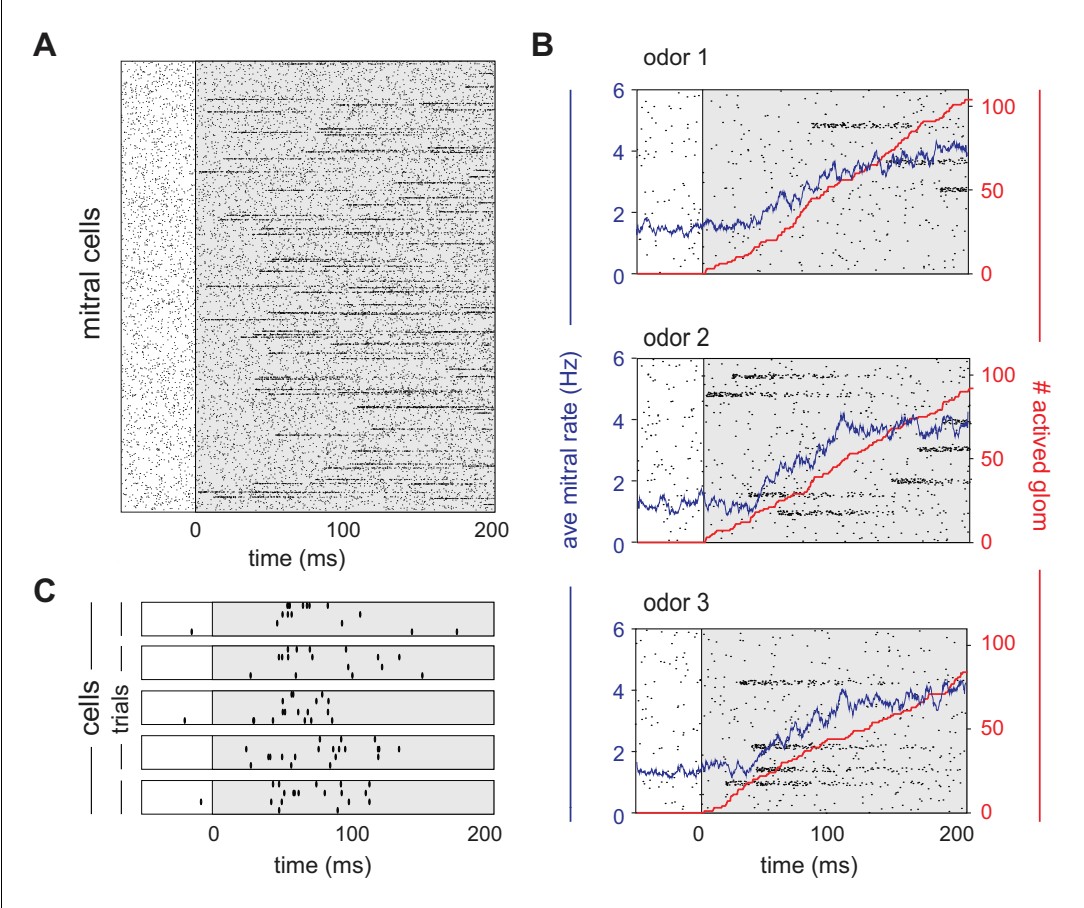

**Figure 2.** Mitral cells are activated with odor-specific latencies. (A) Example raster plot showing all 22,500 model mitral cells (900 glomeruli with 25 mitral cells each) for one odor trial. Each row represents a single mitral cell and all mitral cells belonging to each glomerulus are clustered. Tick marks indicate spike times. Inhalation begins at 0 ms and is indicated by the grey shaded region. (B) Raster plots showing spiking of 1000 mitral cells (40 glomeruli) in response to three different odors. The red curve shows the cumulative number of glomeruli activated across the sniff, and the blue curve is the firing rate averaged across all mitral cells. (C) Raster plots showing trial-to-trial variability for five mitral cells from the same glomerulus in response to repeated presentations of the same odor. Each box represents a different mitral cell, with trials 1–4 represented by the rows within each box.
DOI: https://doi.org/10.7554/eLife.34831.003

specific latencies after the onset of inhalation (*Figure 2A*). Once activated, the firing rates of all model mitral cells associated with that glomerulus step from baseline (1-2 Hz, *Kollo et al., 2014*) to 100 Hz and then decay with a time constant of 50 ms (*Figure 2B*). The spiking of each MTC is governed by a Poisson process (*Figure 2C*). At our reference odor concentration, 10% of the glomeruli are typically activated during the 200 ms sniff.

We modeled a patch of PCx, with connection probabilities and topographies that approximate those characterized in the rodent (*Bekkers and Suzuki, 2013*). The PCx model contains 10,000 excitatory pyramidal cells, each of which receives 50 excitatory inputs from a random subset of the mitral cells and 1000 recurrent excitatory inputs from a random subset of other pyramidal cells (*Figure 3A*). Our model also includes 1225 feedforward inhibitory neurons (FFINs) that receive input from mitral cells and provide synaptic inhibition onto the pyramidal cells and other feedforward interneurons, and a separate population of 1225 feedback inhibitory neurons (FBINs) that each receive inputs from a random subset of pyramidal cells and provide inhibitory input locally onto pyramidal cells and other feedback interneurons. We model all three classes of PCx neurons as leaky integrate-and-fire neurons with current-based synaptic inputs. Model parameter values were constrained wherever possible by the literature and are described in detail in the

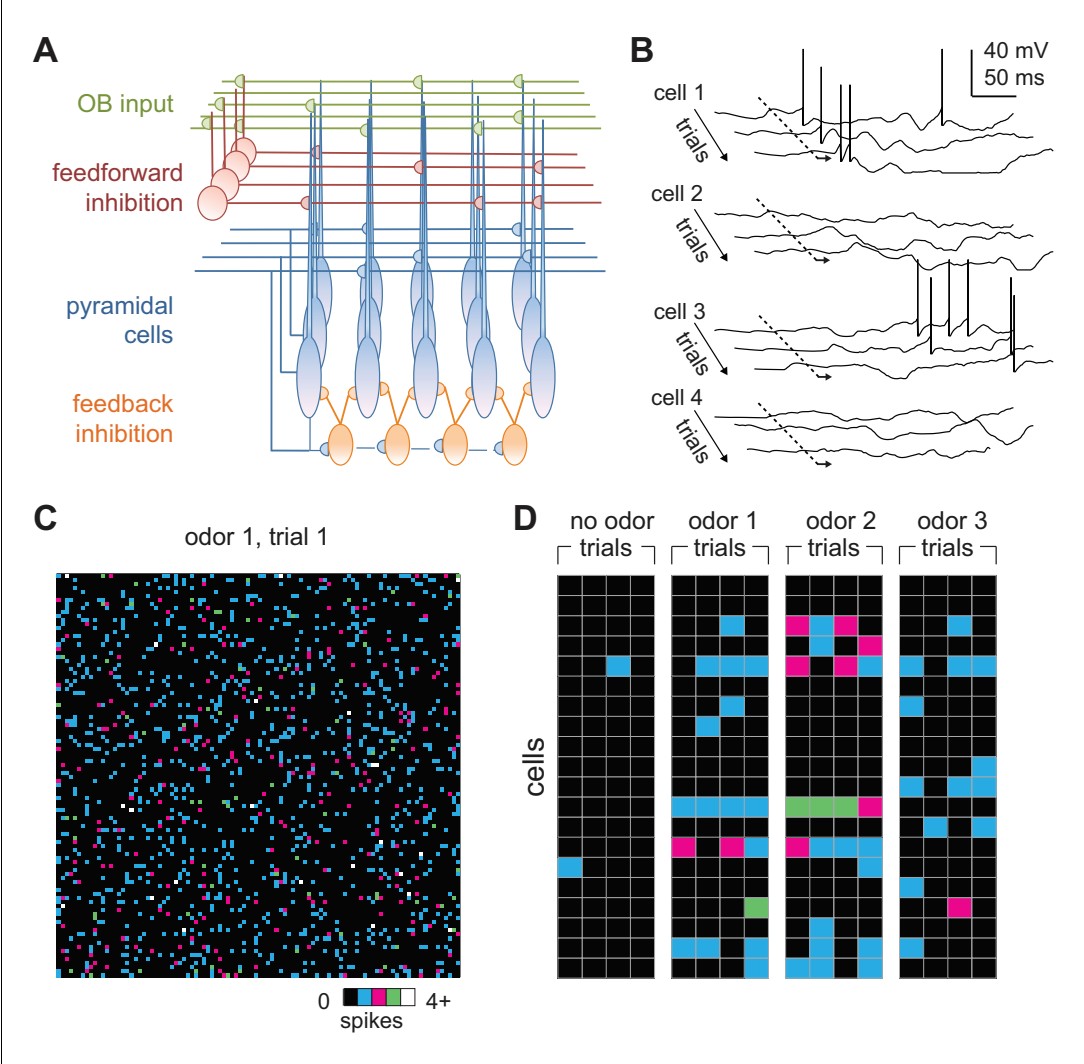

**Figure 3.** Odors activate distributed ensembles of PCx neurons. (A) Schematic of the PCx model. (B) Voltage traces for three sequential sniffs in four model pyramidal cells. Time of inhalation is indicated by the dashed line. (C) Single-trial population activity map for all 10,000 pyramidal cells. Each pixel represents a single cell, and pixel color indicates the number of spikes fired during the 200 ms inhalation. Approximately 13% of cells fired at least one action potential, with activated cells randomly distributed across the cortex. (D) Response vectors shown for 20 cells in response to different odors presented on four sequential trials. Spiking levels are low for no-odor controls. Note the trial-to-trial variability and that individual cells can be activated by different odors.

DOI: https://doi.org/10.7554/eLife.34831.004

Materials and methods. Most of our analyses focus on pyramidal cell activity because these cells receive bulb input and provide cortical output and thus carry the cortical odor code.

Low levels of spontaneous PCx spiking in the model are driven by baseline activity in mitral cells, and 2.8 ± 0.4% (mean ± st. dev) of pyramidal cells spike during the 200 ms inhalation in the absence of odor, consistent with spontaneous firing observed in anesthetized rats (*Poo and Isaacson, 2009*) and near, but slightly lower than, spontaneous rates in awake animals (*Zhan and Luo, 2010*; *Miura et al., 2012*; *Bolding and Franks, 2017*; *Iurilli and Datta, 2017*). We defined any cells that fire at least one action potential during the 200 ms inhalation as 'activated'. Given the low spontaneous firing rates, there was no odor-evoked suppression of firing in the model. Because each piriform cell receives input from a random subset of mitral cells, different odors selectively and specifically activate distinct subsets of pyramidal cells (*Figure 3B*) so that each cell is responsive to multiple odors and each odor activates distinct ensembles of neurons distributed across PCx (*Figure 3C*). At our reference concentration, for which 10% of glomeruli are activated, 14.1 ± 0.59% (mean ± st.

dev., n = 6 odors) of piriform pyramidal cells fire at least one action potential during a sniff, which is consistent with experimental data (*Poo and Isaacson, 2009*; *Stettler and Axel, 2009*; *Miura et al., 2012*; *Bolding and Franks, 2017*; *Iurilli and Datta, 2017*; *Roland et al., 2017*).

PCx cells can exhibit considerable trial-to-trial variability in response to repeated presentations of the same odor (*Otazu et al., 2015*; *Bolding and Franks, 2017*; *Iurilli and Datta, 2017*; *Roland et al., 2017*). To examine trial-to-trial variability in the model, we quantified responses as vectors of spike counts, one component for each pyramidal cell, either over the full 200 ms inhalation or only the first 50 ms after inhalation onset. We then compared pair-wise correlations between response vectors on either same-odor trials or trials involving different odors. Even though glomerulus onset latencies are identical in all same-odor trials, stochastic mitral cell firing results in considerable trial-to-trial variability (*Figure 3D*). We found correlation coefficients for same-odor trial pairs over the full sniff to be $0.35 \pm 0.010$, mean $\pm$ SD (for multiple same-odor trial pairs using six different odors). Pairs of model PCx responses to different odors, on the other hand, had correlations of $0.11 \pm 0.016$; mean $\pm$ SD (for pairs from the same six odors), which is significantly lower than same-odor trial correlations. Both correlation coefficients are smaller than what has been measured experimentally (0.48-same, 0.38-different, *Bolding and Franks, 2017*); 0.67-same, 0.44-different, *Roland et al., 2017*). A number of factors may contribute to increasing correlations beyond what is seen in the model. Gap junctions between MTCs from the same glomerulus correlate their responses (*Christie et al., 2005*; *Schoppa, 2006*), and this would reduce the variability from what the model produces from independent Poisson processes. PCx contains a small subset of broadly activated cells (*Zhan and Luo, 2010*; *Otazu et al., 2015*; *Bolding and Franks, 2017*; *Roland et al., 2017*) that are likely over-represented in the data, and these increase response correlations to different odorants. Furthermore, although PCx cells can either be odor-activated or odor-suppressed, individual cells mostly retain their response polarity across odors, so that a cell that is activated by one odor is rarely suppressed by other odors, and *vice versa* (*Otazu et al., 2015*; *Bolding and Franks, 2017*), a feature not captured by the model. Finally, the higher correlation values may reflect latent structure in PCx connectivity, either innate or activity-dependent, that increases the correlated activity and is not captured by our model.

## Evolution of cortical odor ensembles

We next examined how spiking activity of the four different classes of neurons (mitral cells, pyramidal cells, FFINs and FBINs) evolve over the course of a single sniff (*Figure 4A*). Preceding inhalation, baseline activity in mitral cells drives low levels of spiking in both pyramidal cells and FFINs. FBINs, which do not receive mitral cell input, show no baseline activity. Shortly after inhalation, inputs from the earliest activated glomeruli initiate a dynamic cascade of cortical activity, characterized by a transient and rapid burst of spiking in a small subset of pyramidal cells that peaks ~40 ms after inhalation onset and is then sharply truncated by the strong and synchronous recruitment of FBINs. Pyramidal cell firing rebounds modestly after the synchronous FBIN response, but then the network settles into a sustained state with somewhat elevated pyramidal cell activity that both drives and is held in check by feedback inhibition (*Figure 4A*). Although more mitral cells respond later in the sniff, cortical population spiking levels are stabilized by slowly increasing activity of FFINs, which cancels the increase in total mitral cell input. This rapid and transient increase in pyramidal cells firing followed by sustained cortical suppression despite continued input from olfactory bulb resembles the population spiking patterns we observed experimentally (*Figure 1*).

What triggers the rapid transient pyramidal cell response? Each odor initially activates a group of glomeruli that project randomly onto different cortical pyramidal cells. A small subset of pyramidal cells receives enough direct input from short-latency mitral cells to reach threshold and start spiking early in the sniff (*Figure 4B*, cell 1). This activity produces a small amount of recurrent excitation that is dispersed across the cortex via the long-range recurrent collateral connections. The resulting recurrent excitation can recruit other pyramidal cells that receive moderate but subthreshold OB input (*Figure 4B*, cell 2). However, by itself, recurrent excitation is not strong enough to drive spiking in pyramidal cells that received weak OB input, including from spontaneously active MTCs (*Figure 4B*, cell 3). Consequently, more pyramidal cells will be activated selectively, resulting in even stronger recurrent excitation. The result is a regenerative increase in total pyramidal cell activity and recurrent excitation. However, recurrent excitation onto FBINs is stronger than onto other pyramidal cells (*Stokes and Isaacson, 2010*; *Suzuki and Bekkers, 2012*) so that FBINs are recruited before

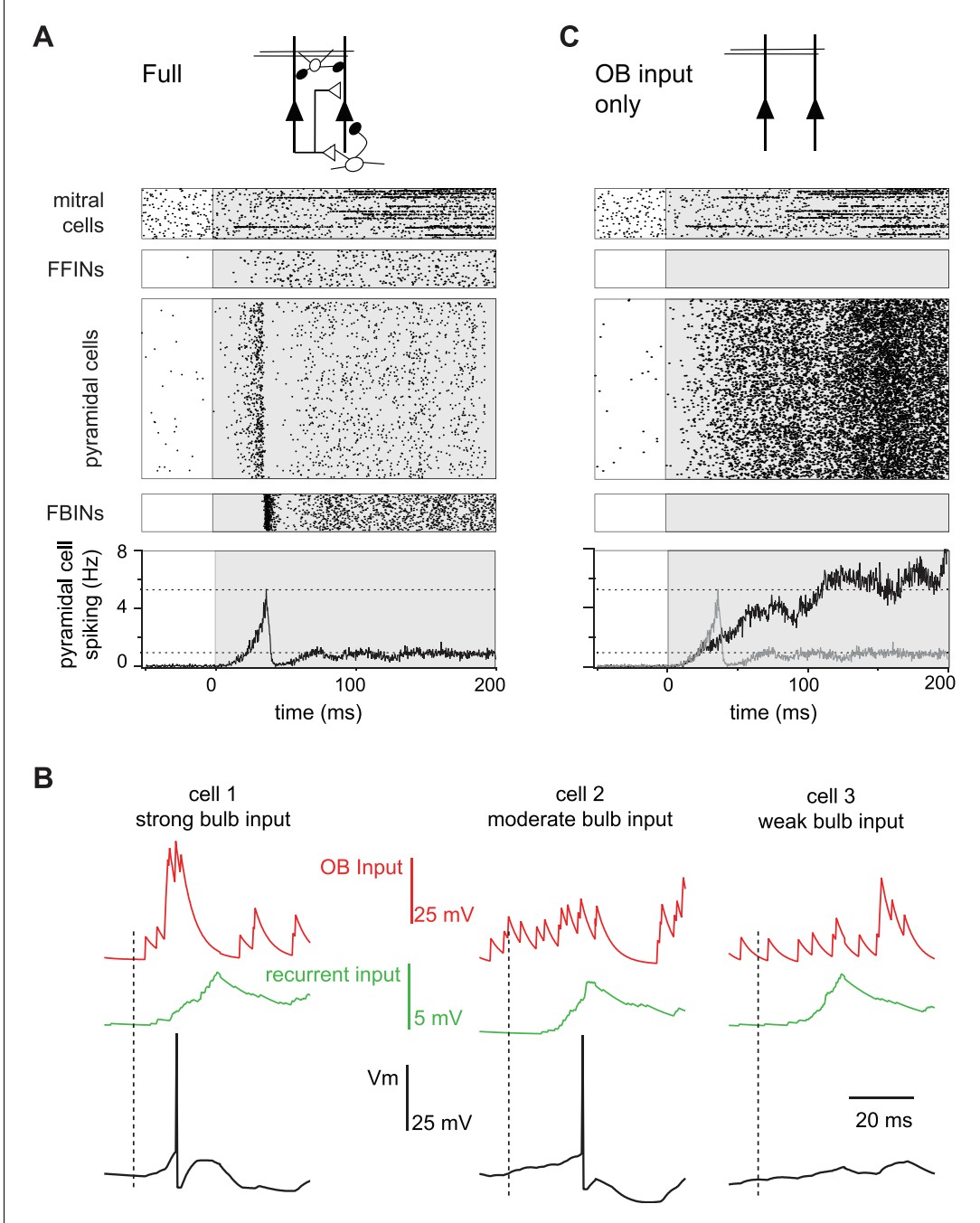

**Figure 4.** Evolution of a cortical odor response. (A) Raster for a single sniff showing spiking activity of a subset of mitral cells (2250 out of 22,500), all 1225 feedforward neurons (FFINs), all 10,000 pyramidal cells, and all 1225 feedback interneurons (FBINs). Spiking rate for the population of pyramidal cells is shown at the bottom (average of six trials). Note that the earliest activated glomeruli initiate a cascade of pyramidal cell spiking that peaks after ~40 ms and is abruptly truncated by synchronous spiking of FBINs. Dashed lines show peak and steady-state firing rates during inhalation. (B) Single-trial voltage traces (black) for three pyramidal cells in response to the same odor. Inhalation onset is indicated by the dashed line. The red traces show OB input and the green traces the recurrent input received by each cell. Cell 1receives strong OB input and spikes soon after odor presentation. Cell 2 receives subthreshold input from OB and only spikes after receiving addition recurrent input from other pyramidal cells. Cell 3 receives no early odor-evoked input from the bulb, and its recurrent input is subthreshold, so it does not spike over the time period shown. (C) Raster plots for a reduced model in which pyramidal cells only get excitatory input from the OB, without FFI, recurrent excitation or FBI. Pyramidal cell spiking tracks mitral cell input. Population rate for the full network is shown in grey for comparison.

DOI: https://doi.org/10.7554/eLife.34831.005

recurrent excitation alone can activate pyramidal cells that only received weak OB input. Thus, feedback inhibition quickly halts the explosive growth of pyramidal cell firing. Because pure recurrent input always remains subthreshold for pyramidal cells, the odor-specificity of the cortical ensemble is maintained.

## Specific roles for different circuit elements in shaping cortical responses

We sought to reveal the specific roles that different circuit elements play in shaping PCx output and to examine the sensitivity/robustness of our model to changes in its parameters. In these studies, the same odor stimulus was used in all cases, so input from the olfactory bulb is identical except for the trial-to-trial stochasticity of mitral cell spiking. We first compared responses in the full circuit (*Figure 4A*) with those in a purely feedforward network in which pyramidal cells only receive mitral cell input (*Figure 4C*). Two key features of PCx response dynamics are different in this highly reduced circuit: first, pyramidal cell spiking increases continuously over the course of the sniff as more glomeruli are activated (*Figure 2B*); second, the strong initial transient peak in population spiking is lost in the purely feedforward circuit. Intracortical circuitry must therefore implement these features of the population response.

We next varied relevant parameters of the intracortical circuitry to determine the role each element of the circuit plays in shaping output. Simply adding FFI to the reduced circuit did not restore the shape of the population response, indicating that FFI does not selectively suppress later PCx activity (*Figure 5—figure supplement 1A*). Instead, FFI modulates the peak of the population response in the full circuit (*Figure 5A*). We observe subtle differences, such as more variable pyramidal activity, if we change the strength of the excitatory OB input onto FFINs rather than the FFI itself (*Figure 5—figure supplement 1B,C*). FFI inhibits both pyramidal cells and FFINs and hence enables the overall amount of inhibition received by pyramidal cells to remain steady across a range of FFI strengths. As the strength of inhibition onto pyramidal cells from a single FFIN increases the recurrent inhibition onto other FFINs increases as well, leading to less active FFINs and hence steady overall inhibition onto pyramidal cells.

Next, we examined responses when we varied FBI (*Figure 5B,C*). Runaway excitation occurs when FBI is significantly weakened (magenta traces, illustrated also in *Figure 5—figure supplement 1D*). Pyramidal cell activity is robust over a large range of FBI values. This is because FBI goes both onto pyramidal cells and other FBINs (via local recurrent inhibitory connections). Similar to FFI, decreasing FBI results in more active FBINs, ultimately resulting in similar total levels of feedback inhibition onto pyramidal cells (*Figure 5Ci*). Increasing the strength of FBI produces a transient decrease in both the number of active pyramidal cells and active FBIN, again, resulting in similar overall feedback inhibition and pyramidal cells activity. However, unlike FFI, this activity is modulated by oscillations due to the feedback circuit, as the FBINs recruited by pyramidal cells are silenced by the strong inhibition that the recruited FBINs themselves produce (*Figure 5Ciii*). Thus, total model output is quite robust to the strength of FBIN inhibition, but population spiking becomes oscillatory when this coupling is strongly increased.

Finally, we examined how model output depends on recurrent excitation. We first examined odor responses when the strength of recurrent excitation onto pyramidal cells and FBINs were co-varied. Total network activity decreased substantially as recurrent excitation strength increased (*Figure 6A*), indicating that FBI overrides pyramidal cell recruitment. Although increasing recurrent excitation did not markedly alter overall response dynamics, both the latency and amplitude of the initial peak decreased with stronger recurrent excitation. By contrast, substantially weakening recurrent excitation produced slow, prolonged and more variable responses. Thus, recurrent excitation is responsible for both the early amplification and the subsequent, rapid truncation of the population response. We next examined the effects of changing recurrent excitation onto either pyramidal cells or FBINs independently (*Figure 6B*). The upward slope to the peak is enhanced by recurrent excitation onto the pyramidal cells, indicating that indeed recurrent excitation is responsible for the recruitment, amplification and rise of pyramidal activity. Accordingly, an increase in its strength gives a higher and earlier peak (*Figure 6Bi*). In contrast, the recurrent excitation onto FBINs modulates the downward slope of the initial peak, as expected for the circuit component responsible for recruiting the inhibition that truncates pyramidal cells activity. Accordingly, an increase in its strength gives an earlier and lower peak (*Figure 6Bii*).

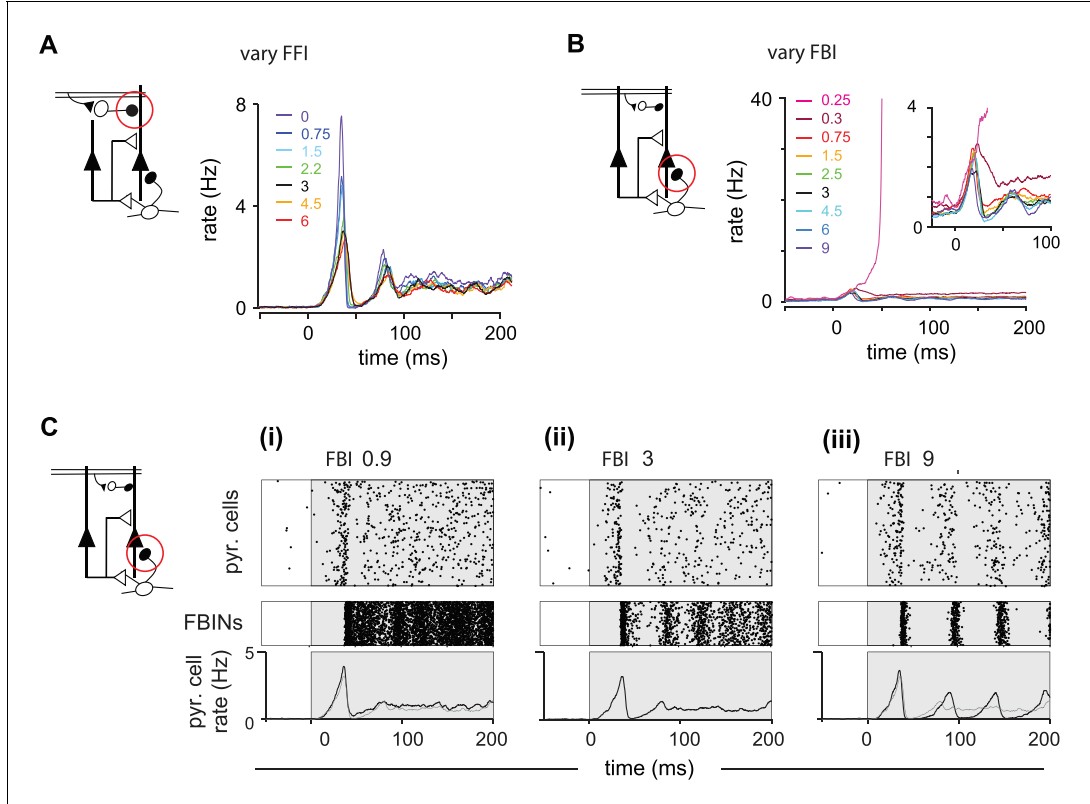

**Figure 5.** Inhibition shapes pyramidal cell spiking. Model output expressed by pyramidal cell population firing rates for multiple parameter values. The varied parameter is indicated by the red circle in the circuit schematics on left. Each colored trace represents the averaged firing rates (six trials each with four different odors). The legend, with colors corresponding to the traces, indicates the peak IPSP for the parameters generating the traces. Black traces show results using default parameter values. (A) Effect of FFI on pyramidal cell output. Different strengths of FFI correspond to peak IPSP amplitudes of 0, 0.75, 1.5, 2.25, 3, 4.5 and 6 mV (see Materials and methods for conversion to parameter values). FFI primarily controls the amplitude of the peak response. (B) Effect of FBI on pyramidal cell output. Different strengths of FBI corresponsd to peak IPSP amplitudes of 0.25, 0.3, 0.75, 1.5, 2.5, 3, 4.5, 6 and 9 mV. Pyramidal cell output is largely robust to changes in the strength of FBI. However, extremely small values of FBI can lead to runaway excitation (see also *Figure 5—figure supplement 1D*). (C) Raster plots for pyramidal cells (showing 3000 cells) and FBINs with different amounts of FBI. (i) Peak IPSP amplitude = 0.9 mV. (ii) Peak IPSP amplitude = 3 mV. (iii) Peak IPSP amplitude = 9 mV. Population spike rates are at bottom, with rates for the control case (ii) overlaid in grey for comparison. While the average pyramidal cell rate is robust to different FBI strength, large values of FBI can lead to oscillations.

DOI: https://doi.org/10.7554/eLife.34831.006

The following figure supplement is available for figure 5:

**Figure supplement 1.** (A–C) Pyramidal cell population firing rates using different parameter values.

DOI: https://doi.org/10.7554/eLife.34831.007

## Piriform responses are shaped by early responding glomeruli

The large and early peak in pyramidal cell spiking suggests that early responding glomeruli play an outsized role in defining the cortical odor response. To examine the relative impact of early- versus late-responding glomeruli directly, we compared the rate of population spiking in our model PCx to the sequential activation of individual glomeruli (*Figure 7A*). In the full network, population spiking peaks 34 ± 8.3 ms after inhalation onset (mean ± SD for six odors with ensemble averages of six trials per odor at the reference concentration; *Figure 7B,C*). At this time, only 15 ± 1.4 glomeruli have been activated out of the 95 ± 6.0 glomeruli that will eventually be activated across the full sniff. In other words, at its peak, PCx activity is driven by the earliest ~15% of activated glomeruli. Mean responses peak slightly earlier when feedforward inhibition is eliminated (28 ± 4.5 ms; *Figure 7B*), with peak activity driven by 12 ± 0.80 glomeruli (*Figure 7B,C*). Population spiking increases much more slowly when recurrent excitation is removed, peaking at 139 ± 29 ms, when most of the responsive glomeruli have been activated (66 ± 0.44; *Figure 7B,C*). Hence, recurrent excitation helps

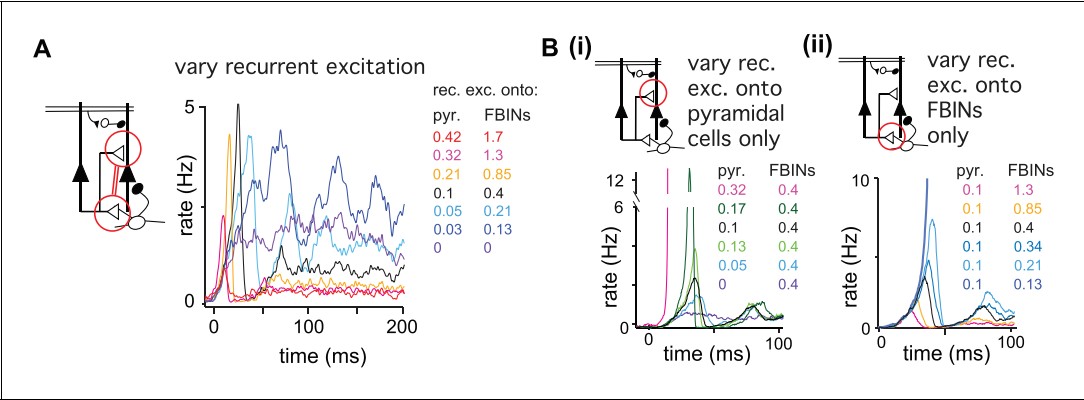

**Figure 6.** Recurrent excitation shapes the early cortical response. Model output expressed by pyramidal cell population firing rates using multiple parameter values. The varied parameters are indicated by the red circle in the circuit schematics. Each colored trace represents the average firing rate (six trials each with four different odors). The legend, with corresponding colors, indicates the maximum values of EPSPs onto pyramidal cells and FBINs. Black traces show results using default parameter values. (**A**) Pyramidal cell population activity with different recurrent collateral couplings. Peak EPSPs onto pyramidal cells of 0, 0.03, 0.05, 0.1, 0.21, 0.32 and 0.42 mV and onto FBINs, 0, 0.13, 0.21, 0.4, 0.85, 1.3 and 1.7 mV. Strong recurrent excitation leads to a stronger initial response but lower activity later in the sniff. Weaker recurrent excitation leads to lower initial response followed by higher and more variable activity. (Bi) Pyramidal cell population activity with different strength recurrent connections onto pyramidal cells only. Peak EPSPs of 0, 0.05, 0.1, 0.13, 0.17 and 0.32 mV. Stronger recurrent connections between pyramidal cells lead to higher and earlier initial response peaks. Even stronger connections lead to runaway pyramidal activity (magenta trace, see also *Figure 5—figure supplement 1D*). (Bii) Pyramidal cell population activity with different recurrent connection strengths onto FBINs only. Peak EPSPs of 0.13, 0.21, 0.34, 0.4, 0.85 and 1.3 mV. Stronger recurrent connections from pyramidal cells onto FBINs lead to lower, yet earlier initial response peaks. Very weak connections lead to runaway activity (purple trace).

DOI: https://doi.org/10.7554/eLife.34831.008

amplify the impact of early-responsive glomeruli and discount the impact of later-responding glomeruli through the recruitment of strong feedback inhibition.

We wondered whether the earliest part of the cortical response provides an especially distinctive representation of odor identity. We therefore compared response correlations over either the full 200 ms inhalation or only the first 50 ms after inhalation onset (see Materials and methods for details). Response correlations to both same-odor and different-odor responses were lower when using only the first 50 ms (same-odor, 0.24 ± 0.019; different-odor pairs, 0.044 ± 0.014; *Figure 7E*). However, the ratio of correlations for same- vs. different-odor responses, which can be thought of as a signal-to-noise ratio, is almost double for responses in the first 50 ms relative to the full 200 ms inhalation (*Figure 7F*). The cortical odor response is therefore largely shaped by the glomeruli that respond earliest in the sniff. Taken together, our model predicts that a cascade of cortical activity is initiated by the earliest-responsive inputs, amplified by recurrent excitation, and then truncated by feedback inhibition, providing a distinctive odor representation.

## Distinct roles for feedforward and feedback inhibition in normalizing PCx output

We next determined how cortical odor representations depend on odorant concentration. Glomerular (*Spors and Grinvald, 2002*) and MTC onset latencies decrease with increasing concentrations of odorant (*Cang and Isaacson, 2003*; *Junek et al., 2010*; *Fukunaga et al., 2012*; *Sirotin et al., 2015*). We simulate this in our OB model by scaling the onset latencies from those at the reference concentration (*Figure 8A*). In other words, to decrease odor concentration, we uniformly stretch latencies, causing fewer glomeruli to be activated within 200 ms, and making those that are activated respond later. Conversely, we shrink the set of latencies to simulate higher concentrations so that glomeruli that were activated later in the sniff at lower concentrations are activated earlier, and some glomeruli that were not activated at lower concentrations become activated at the end of the sniff at higher concentrations. Importantly, stretching or shrinking latencies does not change the sequence in which glomeruli become activated. We quantify odor concentration using the fraction of activated glomeruli. Note that given the nonlinear concentration-dependence of receptor activation

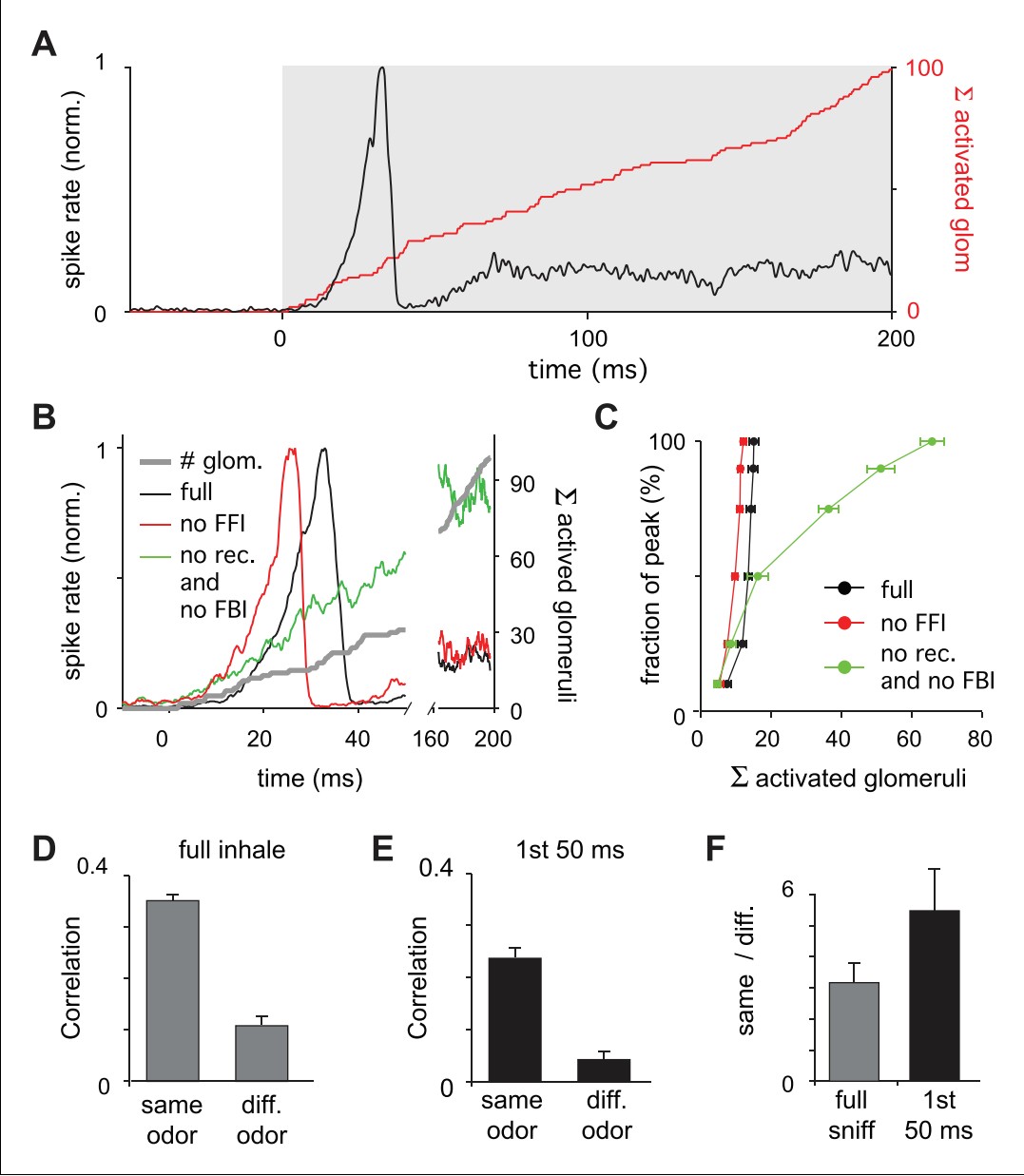

**Figure 7.** Earliest-active glomeruli define the PCx response. (**A**) Normalized population spike rates (black) in response to an odor during the sniff cycle (inhalation indicated by grey background). The red curve shows the cumulative number of glomeruli activated across the sniff. Note that population spiking peaks after only a small subset of glomeruli have been activated. (**B**) Normalized population spike rates for one odor for the full network (black trace), without FFI (red trace) and without recurrent excitation (green trace). Grey trace shows the cumulative number of activated glomeruli. (**C**) Fraction of the peak population spike rate as a function of the cumulative number of activated glomeruli for six different odors. These curves indicate the central role recurrent excitation plays in amplifying the impact of early responsive glomeruli. (**D**) Average correlation coefficients for repeated same-odor trials and pairs of different-odor trials measured over the full 200 ms inhalation. (**E**) As in D but measured over the first 50 ms after inhalation onset. (**F**) Ratios of correlations for same- vs. different-odor trials measured over the full sniff (grey bar on left) and over the first 50 ms (black bar on right).
DOI: https://doi.org/10.7554/eLife.34831.009

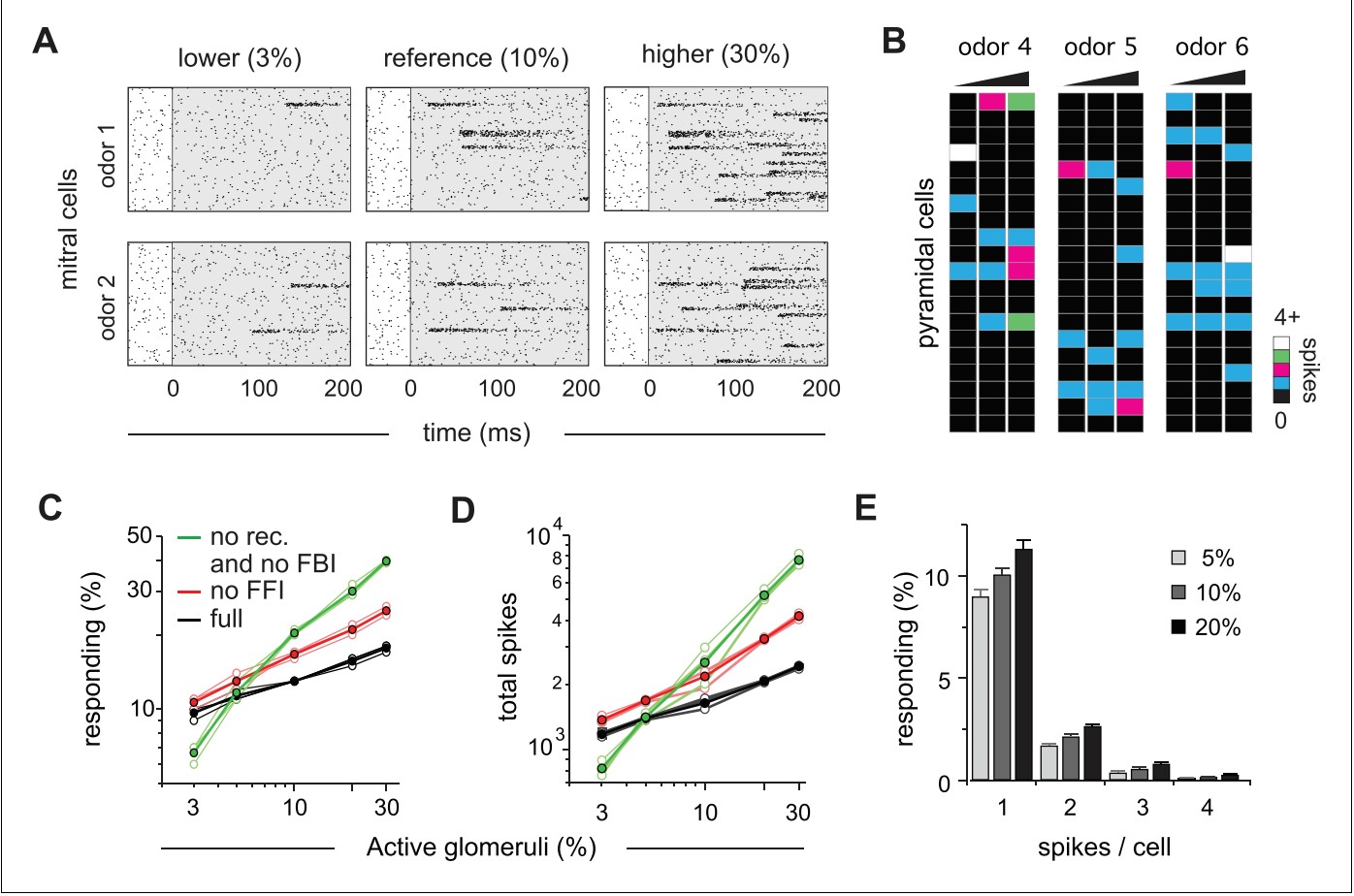

**Figure 8.** Cortical output is normalized across concentrations. (**A**) Mitral cell raster plots for 2 odors at three different concentrations, defined by the fraction of active glomeruli during a sniff. Odors are different from the odors in *Figure 1*. (**B**) Single-trial piriform response vectors over a concentration range corresponding to 3, 10% and 30% active glomeruli. Note that activity does not dramatically increase despite the 10-fold increase in input. (**C**) Fraction of activated pyramidal cells at different odor concentrations for the full network (black trace), without FFI (red trace) and without recurrent excitation (green trace) for four different odors (open circles, thin lines) and averaged across odors (filled circles, thicker lines). Note that eliminating FFI primarily shifts the number of responsive cells, indicating that FFI is largely subtractive, whereas eliminating recurrent excitation alters the gain of the response. Note also that recurrent excitation amplifies the number of activated cells at low-odor concentrations. (**D**) As in C but for the total number of spikes across the population. (**E**) Distribution of spike counts per cell at different odor concentrations. Data represent mean ± s.e.m. for n = 4 odors at each concentration.

DOI: https://doi.org/10.7554/eLife.34831.010

and normalization at multiple stages of the system upstream of the cortex (*Cleland et al., 2011*), a 10-fold increase in mitral cell output corresponds to a much greater range of concentrations.

The number of responsive pyramidal cells is buffered against changes in odor concentration (*Figure 8B*). Across the population, we found that the number of responsive pyramidal cells only increases by 80% upon a 10-fold increase in input (mean ±s.d.; 9.7 ± 0.40% of pyramidal cells respond when 3% of glomeruli are active; 17.3 ± 0.71% of pyramidal cells respond when 30% of glomeruli are active; *Figure 8C*). This indicates that the size of cortical odor ensembles is only weakly concentration-dependent, which is consistent with experimental observations (*Stettler and Axel, 2009*; *Bolding and Franks, 2017*; *Roland et al., 2017*). In addition, both the total number of spikes across the population (*Figure 8D*) and the number of spikes evoked per responsive cell (*Figure 8E*) are only modestly, but uniformly, concentration-dependent. Recent imaging studies indicate that subsets of piriform cells are especially robust to changes in concentration (*Roland et al., 2017*). It is not yet known how this subset of cells emerges in PCx, and this result is not recapitulated in our model where all cells are qualitatively similar in terms of input, intrinsic properties and local

connectivity. Note that we are simulating a situation in which OB output scales very steeply with concentration. In fact, considerable normalization across concentrations occurs within OB (*Cleland et al., 2011*; *Banerjee et al., 2015*; *Sirotin et al., 2015*; *Roland et al., 2016*; *Bolding and Franks, 2017*). Nevertheless, this normalization is incomplete. Our model now shows that a relatively simple PCx-like circuit is sufficient to implement this normalization.

To gain insight into how normalization is implemented, we again simulated responses at different concentrations, but now either without FFI or without recurrent excitation and FBI. Eliminating FFI increases both the number of responsive cells (*Figure 8C*) and total population spiking (*Figure 8D*). However, this increase is fairly modest, uniform across concentrations, and does not substantially change the gain of the response (i.e. the slope of the input-output function). This indicates that the effect of FFI is largely subtractive, consistent with our earlier analysis (*Figure 5*). In marked contrast, responses become steeply concentration-dependent after eliminating recurrent excitation and FBI, dramatically increasing response gain. Interestingly, cortical output is reduced at low odor concentrations when recurrent excitatory and FBI are removed, indicating that recurrent collateral excitation also amplifies cortical output in response to weak input (*Figure 8C,D*). Thus, our model demonstrates that a recurrent, piriform-like circuit bi-directionally normalizes graded input by amplifying low levels of activity via recurrent collateral excitation between pyramidal cells and suppressing high levels of activity by recruiting scaled FBI.

## Early activated PCx cells support concentration-invariant odor decoding

We quantified response similarity, using spike counts over the full 200 ms inhalation. To do this, we calculated response correlations to an odor at our reference concentration (10% active glomeruli) and compared these to either responses to the same odor (*Figure 9A*, black curve) or different odors (*Figure 9A*, blue curve) at different concentrations. Responses to the same odor became more dissimilar (i.e. response correlations decreased) as the differences in concentration increased. By contrast, although responses to different odors were markedly dissimilar (i.e. much lower correlations), these did not depend on concentration. This means that responses to other odors remain more different than same odor responses across concentrations, which could support discriminating between different odors across concentrations. However, these differences become less pronounced at the lowest and highest concentrations.

We next asked if a downstream observer can reliably identify an odor using population spiking, and whether the same odor can be recognized when presented at different concentrations. To do this we trained a readout to identify a specific odor at one concentration (10% active glomeruli) and then asked how well it can distinguish that odor from other odors and how well it can identify the trained odor when it is presented at different concentrations (see Materials and methods for details). We first used spike counts over the full 200 ms inhalation as input. Classification was excellent when trained and tested at a single concentration indicating that, despite considerable trial-to-trial variability (*Figure 2D*), responses to different odors can be distinguished reliably (*Figure 9B*). We then examined classifier performance when tested on different concentrations without retraining. Consistent with the differences in response correlations, performance was excellent around the training concentration but fell off steeply at the lowest and highest concentrations.

Because the sequence of glomerular activation latencies is preserved across concentrations, with the highest affinity glomeruli for a given odorant always activated first, we suspected that the earliest activated glomeruli could provide a more concentration-invariant odor representation of odor identity than the full 200 ms response. To test this prediction, we analyzed early responses by examining spike counts over just the first 50 ms after inhalation. Correlations over first 50 ms were substantially lower than those for the full 200 ms inhale: this was the case for both repeated presentations of the same odor (*Figure 9A*, magenta curve) as well as for responses to different odors (*Figure 9A*, red curve). However, as noted previously (*Figure 7F*), decreasing both sets of correlations increases the ratio of same-odor versus different-odor correlations. Indeed, responses within the first 50 ms contained sufficient information remained for accurate decoding (*Figure 9B*). And, in contrast to full-inhale responses, classification was not only excellent at and near the training concentration, but across all concentrations tested. This occurs because responses remained similar across concentrations at concentrations above the reference (i.e. response correlations were unchanged), which was not the case with the full, 200 ms responses. Thus, the first 50 ms spike count correlations leave a margin between same and different odor responses across all concentrations,

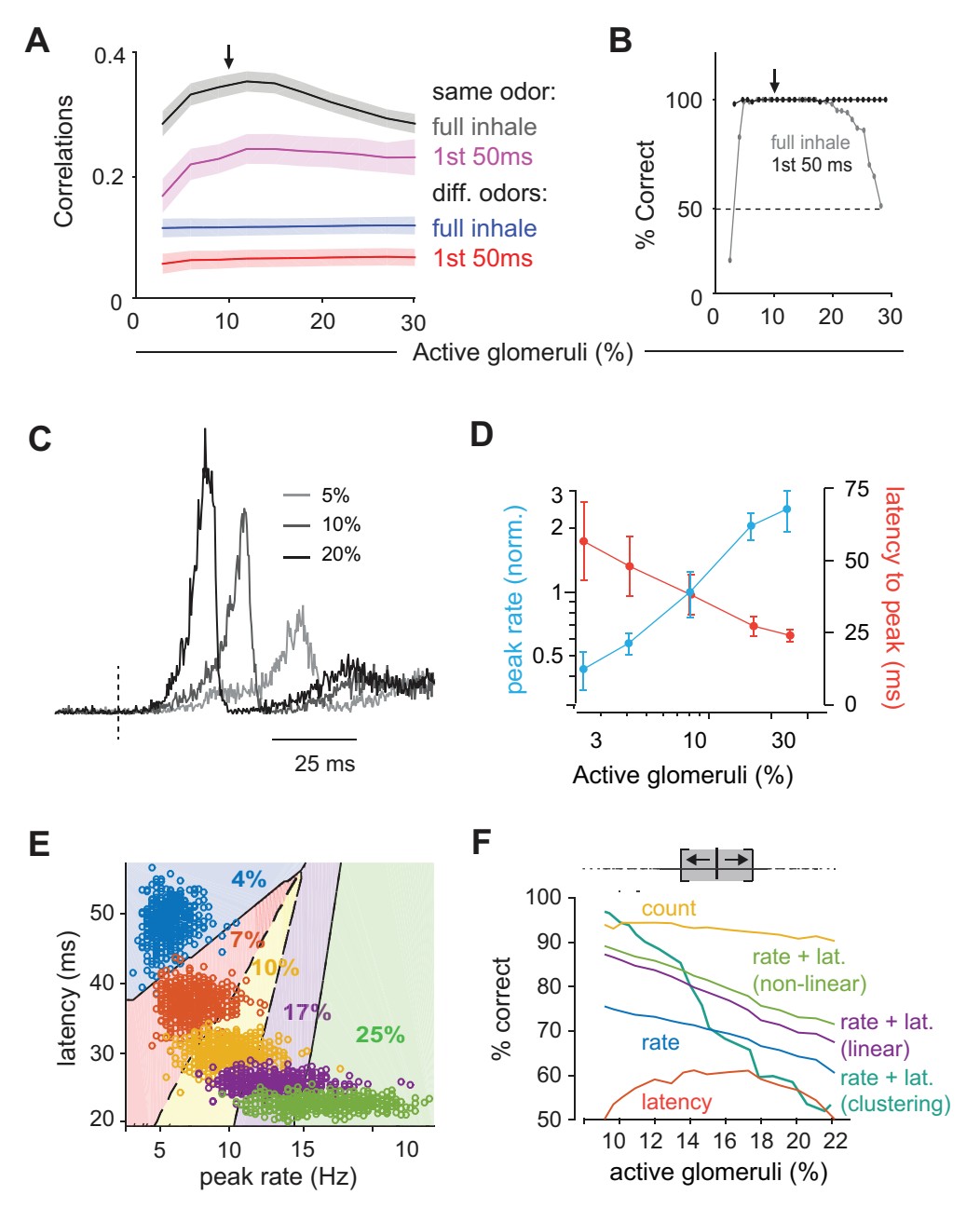

**Figure 9.** Coding of odor identity and concentration. (**A**) Correlation coefficients between responses of a target odor with 10% active glomeruli (black arrow) and the same (black and pink curves) or different (blue and red curves) odors across concentrations. Correlations were calculated using pyramidal cell activity from the full inhale (black and blue curves) or from the first 50 ms of inhalation (pink and red curves). For correlations with the same odor, 25 trial with 10% active glomeruli were paired with 25 trials at each different concentration. For correlations with other odors, 100 trials with the target odor at 10% active glomeruli were paired with each of the 100 other odors at each different concentration. Lines show the mean result and shaded areas show the standard deviation. (**B**) Readout classifications of odor identity when presented at different concentrations. Either the transient cortical activity (first 50 ms of the inhalation; black curve) or the activity across the full inhalation (gray curve) was used for both training and testing. Training was performed solely at the reference concentration (black arrow). The dashed line shows the chance level of classification. (**C**). Example of population spike rates for an odor at three concentrations. Response amplitudes are normalized to the responses at the highest concentration. Dashed lines indicate inhalation onset. (**D**) Average peak firing rate (blue) and latencies to peak (orange) of the population response vs. number of activated glomeruli (four odors). (**E**) Distribution of peak latencies and firing rates for one

*Figure 9 continued*

odor presented at five concentrations. Different colors represent distinct concentrations (fraction of active glomeruli). Background colors indicate classification into one of 5 concentrations (with clustering method). (F) Concentration classification accuracy using different features of the population response. (top) For each target concentration, responses within a ± 3% range (grey shading) were presented and classified as lower or higher than the target. Different features of the population response and techniques used for classification (see Materials and methods) are indicated by colored lines. Dashed lines in B indicate classification boundaries for the clustering classifier using rate + latency.

DOI: https://doi.org/10.7554/eLife.34831.011

supporting the idea that the earliest cortical response can support concentration-invariant odor recognition (*Hopfield, 1995*; *Schaefer and Margrie, 2012*).

## Encoding odor intensity using population synchrony

Finally, we asked how odor intensity could be represented in PCx. To that end, we examined the dynamics of population spiking in response to odors at different concentrations (*Figure 9C*). The peak amplitude of the population response in our PCx model increases substantially at higher concentrations: a 10-fold increase in active glomeruli (3% to 30%) produces a 5.7-fold increase in peak spike rate (*Figure 9D*). However, the same concentration range produced a much smaller increase in the number of responsive cells (1.8-fold, *Figure 8C*) and total spikes (2.1-fold, *Figure 8D*), indicating that population synchrony is especially sensitive to concentration. Response latencies also decrease at higher concentrations (*Figure 9D*). These data suggest that either the population spike count, population synchrony or amplitude, timing, or a combination of these, could be used to represent odor concentration.

We again used a decoding analysis to test this hypothesis (see Materials and methods for details). For a given odor we simulated 500 presentations at each concentration, across a range of concentrations. We then trained a classifier to distinguish between responses to concentrations corresponding to ±3% active glomeruli above or below the target concentration (*Figure 9E*), and quantified classification performance with cross-validation. We used peak rate or latency features of the full population peak response for decoding. Performance was better using the peak rate than latency to peak, and even better when we used a combination of rate and latency. Performance improved marginally using a nonlinear (log) decoder. We also decoded using non-parametric clustering (Materials and methods), which performed almost perfectly at low concentrations, but performance deteriorated as concentration increased. Response timing is more variable as concentration is increased (*Figure 9E*), making it harder to decode based on similarity at large concentrations. Finally, although PCx response rates are buffered, they are not completely insensitive to concentration (a 10-fold increase in OB input results in only a 78% increase in PCx output). Because of their relatively low variability, spike counts can be used for effective concentration classification in our model. Thus, our data suggest that distinct intensity coding strategies may be optimal at different concentrations. However, as noted above, substantial normalization occurs upstream of PCx and total PCx spiking output does not increase with concentration, indicating that spike count is unlikely to be used to encode odor intensity in PCx. Instead, an 'ensemble-identity'/'temporal-intensity' coding strategy has recently been observed in PCx in awake mice (*Bolding and Franks, 2017*). Our model shows how this multiplexed coding strategy can be implemented in a recurrent circuit with the general properties of the PCx.

## Discussion

We sought to understand how temporally structured odor information in the OB is transformed in the PCx. A previous study (*Sanders et al., 2014*) proposed a general scheme for transforming latency codes into ensemble codes, but this model was incompatible with PCx circuitry. We simulated odor-evoked spiking in the OB and used it as input to a PCx network model of leaky integrate-and-fire neurons. Other computational studies have examined how PCx can support oscillatory activity (*Wilson and Bower, 1992*; *Ketchum and Haberly, 1993*; *Protopapas and Bower, 1998*) or auto-associative memory formation (*Barkai et al., 1994*; *Hasselmo and Barkai, 1995*; *Kilborn et al.,*

1996; *Haberly, 2001*); we have not attempted to address these issues. Instead, we show how a PCx-like circuit is sufficient to broadly recapitulate experimental observations, including ensemble codes for odor identity, normalization across odor concentrations, and temporal codes for odor intensity. In doing so, our model provides mechanistic insight into the circuit operations that implement the transformation from a temporal to an ensemble code for odor identity.

A given odor typically activates ~10% of neurons distributed across PCx (*Poo and Isaacson, 2009*; *Stettler and Axel, 2009*; *Miura et al., 2012*; *Bolding and Franks, 2017*; *Roland et al., 2017*). In brain slices, PCx principal cells (pyramidal and semilunar) require multiple (~6) co-active MTC inputs to reach spike threshold (*Franks and Isaacson, 2006*; *Suzuki and Bekkers, 2006*). Our model shows that only a small subset of the total ensemble of responsive PCx neurons need to receive supra-threshold OB input. Because pyramidal cells are connected via long-range recurrent collateral inputs, the few cells that are directly activated by early OB inputs provide diffuse excitatory synaptic input to other cells across PCx. This recurrent excitation brings a larger subset of cells that received moderate, but still subthreshold OB input to spike threshold. This cascade of cortical activity continues until FBINs, which do not receive OB input, are activated. Once activated, FBINs strongly suppress subsequent cortical spiking. This mechanism ensures that the earliest activated glomeruli largely define cortical odor ensembles.

## Subtractive versus divisive inhibition

Whether and why different types of GABAergic inhibition have subtractive or divisive effects is currently an area of intense interest, including in PCx (Isaacson and Scanziani, 2011). Differences in these types of operations are thought to depend on the types of inhibitory interneurons (e.g. SOM vs. PV cells) and their target sites on the postsynaptic cell (i.e. dendrite- vs. soma-targeting). For example, *Sturgill and Isaacson (2015)* recently showed that SOM-mediated inhibition in PCx is almost completely subtractive while PV-mediated inhibition is largely divisive. Our model has two types of inhibition, FFI and FBI, that differ from each other only by their connectivity (i.e. place within the circuit) and are otherwise implemented in the same way. Nevertheless, in our model, FFI and FBI play very different roles in transforming OB input, suggesting that the circuit motif to which an inhibitory neuron belongs determines its role, whereas the inhibitory cell type may only have a secondary impact. In particular, we showed that the slope of the population input-output relationship (i.e. gain) is steeper when recurrent excitation/FBI is removed (leaving only FFI), indicating that recurrent excitation/FBI effectively controls gain while FFI is relatively ineffective at doing so (*Figure 8*). In contrast, gain barely changes when FFI is removed (leaving recurrent excitation/FBI), indicating that FFI's contribution is predominantly subtractive. This result is different from many models for divisive normalization in sensory systems in which the implementation is through feedforward inhibition (*Carandini and Heeger, 2012*). This difference may reflect the fact that the period during which the stimulus arrives in the cortex (i.e. the duration of the sniff, here 200 ms) is much longer than the membrane time constant (20 ms) and the time course of synaptic inhibition (10 ms). Thus, a circuit motif in which recurrent excitation drives strong, scaled feedback inhibition may be better suited to normalizing representations in structures that use temporal or latency-based codes, as opposed to those using more instantaneous, rate-code-based inputs.

## Experimental predictions

Our model makes a number of experimentally testable predictions. PCx is a highly recurrent circuit in which broad and non-specific GABAergic blockade invariably results in epileptiform activity. However, our model predicts that selectively blocking FFI should produce an additive increase in response amplitude but not dramatically alter response dynamics. In contrast, selectively and partially blocking FBI should have a large and multiplicative effect. Recent identification of genetic markers for different classes of PCx interneurons (*Suzuki and Bekkers, 2010*) should facilitate these experiments. In fact, different subtypes of PCx FBINs have been reported to have distinct effects on odor responses (*Sturgill and Isaacson, 2015*), a result that would require additional cell-types in our model to explain. Interestingly, even though our model predicts that odor responses will be sensitive to partial blockade of excitatory input onto FBINs, it is highly robust to partial blockade of feedback inhibition.

Piriform pyramidal cells are interconnected by excitatory recurrent collateral connections. Our model makes the somewhat counter-intuitive prediction that reducing pyramidal cell output will substantially increase and prolong the odor response (*Figure 6A*). This prediction is motivated by the much greater impact of FBI than FFI on driving the population response. Moreover, blocking pyramidal cell output should make the usually normalized response steeply concentration-dependent (*Figure 8C,D*). These predictions could be tested, for example, by blocking output using viruses to selectively express tetanus toxin (*Murray et al., 2011*) in PCx pyramidal cells. Our model also suggests that odor intensity could be encoded in temporal features of the population response. While complicated, psychophysical experiments with optogenetic activation of subsets of PCx neurons could provide a way to test this prediction (*Smear et al., 2011*). Additionally, we find lower same-odor response correlations than have been observed experimentally. Independent Poisson spiking in mitral cells provides the major source of trial-to-trial variability in our model. However, sister MTCs are connected through gap junctions and often exhibit highly correlated spiking that is entirely absent in Connexin-36 knock-out mice (*Christie et al., 2005*). Our model predicts that PCx odor responses in Connexin-36 knock-out mice would exhibit more trial-to-trial variability. Finally, pyramidal cells are interconnected randomly in our model. However, this circuitry remains plastic into adulthood (*Poo and Isaacson, 2009*) and is thought to provide a substrate for odor learning and memory (*Haberly, 2001*; *Wilson and Sullivan, 2011*). Selectively interconnecting pyramidal cells that receive common input and are therefore often co-active would decrease trial-to-trial variability. This prediction could be tested, for example, by constitutively eliminating NMDA receptors from pyramidal cells.

## Limitations of our model circuit

*Bolding and Franks, 2017* observed a biphasic population response in PCx in which some responses are rapid and largely concentration-invariant while others occur with longer latencies that decrease systematically with odorant concentration. The data provided in *Figure 1* show similar biphasic responses in both OB and PCx (*Figure 1E*). This feature is not recapitulated in the model for several possible reasons. First, we modeled a single population of MTCs without distinguishing mitral versus tufted cells. In fact, mitral cells have longer response latencies that decrease at higher odor concentrations, while tufted cells have much shorter response latencies (*Fukunaga et al., 2012*). Furthermore, we have not modeled centrifugal projections from PCx back to OB (*Boyd et al., 2012*; *Markopoulos et al., 2012*; *Otazu et al., 2015*). The initial peak in PCx firing could drive transient inhibition in OB, which could produce a biphasic response that would better match our experimental observations (*Figure 1E*).

We did not attempt to model, in either OB or PCx, responses that are suppressed below background by odor (*Shusterman et al., 2011*; *Fukunaga et al., 2012*; *Economo et al., 2016*). We have also not attempted to distinguish between different subclasses of principal neurons (e.g. semilunar cells versus superficial pyramidal cells), different types of inhibitory GABAergic interneurons, or more sophisticated neural circuit motifs, such as disinhibition, which has been observed in PCx (*Sturgill and Isaacson, 2015*; *Large et al., 2016*). We have also only modeled OB and PCx activity over a single respiration cycle. We justify this simplification based on the observation that highly trained rodents can discriminate between odors (*Uchida and Mainen, 2003*; *Abraham et al., 2004*; *Rinberg et al., 2006*) or odor concentrations (*Resulaj and Rinberg, 2015*) within a single sniff, indicating that sufficient information must be encoded within that time to represent these features. Nevertheless, odor responses in OB and PCx exhibit pronounced oscillations at beta and gamma frequencies, and representations can evolve over a period of seconds (*Kay et al., 2009*; *Bathellier et al., 2010*). These dynamics may be important in more challenging and ethologically relevant conditions. While we note that beta-like oscillatory activity can emerge in our PCx model when feedback inhibition is strong (*Figure 5C*), we have not incorporated or examined these dynamics in detail here.

## What information is relevant for cortical odor coding?

The PCx response in our model is dominated by early glomerular input and relatively unaffected by later glomerular activations. Why would a sensory system discard so much information about a stimulus? To respond to a huge variety of odorants, the olfactory system employs a large number of

distinct odorant receptors that each bind to multiple odorants with various affinities. This implies a reduction in OSN selectivity at high concentrations (*Malnic et al., 1999*; *Jiang et al., 2015*). Nevertheless, high-affinity glomeruli will always be activated earliest. By defining cortical odor ensembles according to the earliest responding glomeruli, the olfactory system uses information provided by high-affinity receptors and discounts information provided by less-specific and possibly spurious receptor activations. Trained rodents can identify odors within ~100 ms, well before most responsive glomeruli are activated (*Wesson et al., 2008*), indicating that activation of only the earliest-responding glomeruli conveys sufficient information to PCx to accurately decode odor identity (*Hopfield, 1995*; *Schaefer and Margrie, 2007*; *Schaefer and Margrie, 2012*; *Jiang et al., 2015*; *Wilson et al., 2017*)

Our model shows how a PCx-like recurrent circuit amplifies the impact of the earliest inputs and suppresses impact of those that arrive later. This not only normalizes total spiking output, but also enhances odor recognition across concentrations. In fact, we found that a downstream decoder can more accurately recognize odors across a large concentration range when using only early activity. This occurs, in part, because the full (i.e. 200 ms) representation is corrupted by spontaneous activity at low concentrations and contaminated by inputs from late-responding glomeruli at high concentrations. However, it is important to note that, in the model, the sequential activation of glomeruli across the sniff is fully defined. In reality, activation of lower-affinity glomeruli will be far less specific than higher affinity glomeruli, so that input to PCx output becomes increasingly less odor-specific later in the sniff. Our model therefore likely underestimates the advantage of decoding odor identity using the earliest-activated PCx cells.

In conclusion, we find that a recurrent feedback circuit can implement a type of temporal filtering of information between OB and PCx in which the earliest-active cells in OB have an outsized role in shaping odor representations in PCx. This transformation supports multiplexed representations of odor identity and odor concentration in PCx. Recurrent normalization has been shown to be particularly effective for controlling the gain in other structures that use phasic or time-varying input (*Louie et al., 2014*; *Sato et al., 2016*). Thus, we propose that the transformation of odor information from OB to PCx is an instance of a more widely-implemented circuit motif for interpreting temporally structured input.

## Materials and methods

### Modeling

The model was written in C and compiled using Apple's xcode environment. The model was run as an executable in OS 10.10+. Runtime for a single trial was approximately 1 s. Source code and an executable model are provided as Supplementary files.

### Model olfactory bulb

The model bulb includes 900 glomeruli with 25 model mitral cells assigned to each glomerulus.

For every odor, each glomerulus is assigned a reference onset latency between 0 to 200 ms. The actual glomerular onset latencies for a given concentration are obtained by dividing the set of reference latencies by $f$, the fraction of glomeruli activated at a particular odor concentration (odor concentrations are defined by the value of $f$ used). Glomeruli with latencies longer than the duration of the inhalation, 200 ms, are not activated. At our reference concentration $f_{ref}$ = 10% of the glomeruli have onset latencies < 200 ms. Mitral cell spiking is modeled as a Poisson process that generates action potentials at specified rates; the baseline spike rate is either 1.5 or 2 Hz, this steps to 100 Hz when a glomerulus is activated and then decays back to baseline with a time constant of 50 ms. Poisson-generated mitral cell spiking introduces stochasticity into our olfactory bulb model.

### Model piriform architecture and connectivity

The piriform model includes three types of model cells: 10,000 excitatory pyramidal cells, 1225 feed-forward inhibitory neurons (FFINs), and 1225 feedback inhibitory neurons (FBINs). The model pyramidal cells and FBINs are assigned to locations on a two-layer grid. Pyramidal cells and FBINs are uniformly spread over the grid on their respective layers. Each pyramidal cell receives an input from 1000 other pyramidal cells and from 50 FFINs, both randomly chosen independent of location. Each

pyramidal cell receives local input from the closest 12 (on average) FBINs. Each FBIN receives input from 1000 randomly chosen pyramidal cells and the 8 (on average) closest FBINs. Each FFIN receives input from 50 other randomly chosen FFINs. Each mitral cell sends input to 25 randomly selected cells (either pyramidal cells or FFINs) in the pirifom. As a result, each pyramidal and FFIN receives input from approximately 50 randomly selected mitral cells. Our study focuses on understanding properties of the activity of pyramidal cells because these provide the only output of the piriform cortex. Hence, the connectivity structure is built to replicate the inputs statistics 'seen' by the pyramidal cells, as determined experimentally.

## Piriform dynamics

The piriform cells are modeled as leaky integrate-and-fire neurons with membrane potential (V) of model piriform cell $i$ obeying the dynamical equation

$$\tau_m \frac{dV_i}{dt} = (V_r - V_i) + I_i^{ex} - I_i^{in}$$

Here. $\tau_m = 15$ ms is the membrane time constant, $V_r$ is the resting potential and $I_i^{ex}$ and $I_i^{in}$ are the excitatory and inhibitory synaptic currents, respectively. We have absorbed a factor of the membrane resistance into the definition of the input currents so they are measured in the same units as the membrane potential (mV). FFINs and FBINs have a resting potential of $V_r = -65$ mv. Pyramidal cell resting potentials are taken from a Gaussian distribution with mean $-64.5$ mv and standard deviation 2 mV. When the membrane potential reaches the firing threshold, $V_{th} = -50$ mV, the neuron fires an action potential and the membrane potential is reset to a reset value $V_{reset} = -65$ mV, where it remains for a refractory period $\tau_{ref} = 1$ ms. The membrane potential is clamped when it reaches a minimum value of $V_{min} = -75$ mV.

The excitatory and inhibitory synaptic currents, $I_i^{ex}$ and $I_i^{in}$, decay exponentially to zero with time constants of 20 and 10 ms, respectively. The excitatory current combines two components, AMPA and NMDA, into a single current. Because the NMDA synapses are relatively slow and AMPA relatively fast, we choose the time constant of this composite current in an intermediate range between these extremes.

Each action potential fired by a neuron induces an instantaneous jump in the current of all its postsynaptic targets by an amount equal to the appropriate synaptic strength. Action potentials in FFINs and FBINs affect the inhibitory currents of their postsynaptic target neurons, and action potentials in the pyramidal and mitral cells affect the excitatory currents of their postsynaptic targets. We denote the jump in the synaptic current induced by a single presynaptic action potential by $I$. It is convenient to give, in addition, the peak postsynaptic potential produced by a single action potential, denoted by $V$. For a membrane time constant $\tau_m$ and a synaptic time constant $\tau_s$, the relationship between $I$ and $V$ is $V = I\tau_r(a^b - a^c)/\tau_m$ where $\tau_r = \tau_m\tau_s/(\tau_m - \tau_s)$, $a = \tau_s/\tau_m$, $b = \tau_r/\tau_m$, and $c = \tau_r/\tau_s$. Except where otherwise noted (figure captions), the values of $I$ for excitatory connections from pyramidal-to-pyramidal, pyramidal-to-FBIN, mitral-to-pyramidal and mitral-to-FFIN are 0.25, 1, 10 and 10 mV, respectively, corresponding to $V$ values of 0.1, 0.4, 4 and 4 mV. The values of $I$ for inhibitory connections from FFIN-to-pyramidal, FBIN-to-FBIN, and FBIN-to-FBIN are all $-10$ mV, corresponding to a $V$ value of $-3$ mV.

## Pyramidal cell population activity vectors

To analyze cortical responses, we define an activity vector $\vec{r}$. Each component of $\vec{r}$ is the number of spikes generated by a pyramidal neuron, starting at the beginning of the inhalation. The spike count continues across the full inhale, or stops after 50 ms in cases when we are interested in the initial response only. The activity maps in the *Figures 3D* and *8B* are a visual representation of the activity vectors created by reshaping the vectors and assigning a color on the basis of their component values.

## The readout

We use a readout defined by a weight vector $\vec{w}$ to classify odor responses to bulb input on the basis of the activity vectors explained above. Our goal is to train the readout so that trials involving a chosen target odor are distinguished from trials using all other odors. Because we generate odors

randomly and all model mitral cells behave similarly, the results are independent of the choice of the target odor. Distinguishing the activity for a target odor from all other activity patterns means that we wish to find $\vec{w}$ such that trials with a target odor have $\vec{w} \cdot \vec{r} > 0$ and trials with other odors have $\vec{w} \cdot \vec{r} < 0$. Such a $\vec{w}$ only exists if trials using the target odor are linearly separable from trials using other odors. If such a readout weight vector exists, this indicates that pyramidal cell activity in response to a specific odor is distinguishable from activity for other odors.

During training, 100 odors were presented at a specific concentration (10% activated glomeruli) over a total of 600 trials. Odor one was chosen as the target, and the trials alternated between this target odor and the other odors. Thus, odor 1 was presented 303 times and every other odor three times. On every trial, the quantity $\vec{w} \cdot \vec{r}$ was calculated, with $\vec{r}$ the activity vector for that trial and $\vec{w}$ the current readout weight vector. Initially, $\vec{w}$ was zero. If classification was correct, meaning $\vec{w} \cdot \vec{r} > 0$ for the target odor or $\vec{w} \cdot \vec{r} < 0$ for other odors, $\vec{w}$ was left unchanged. Otherwise $\vec{w}$ was updated to $\vec{w} + \vec{r}$ or $\vec{w} - \vec{r}$ for trials of odor one or for other odors, respectively. The entire training procedure was repeated twice, once with activity vectors that included spikes counts around the peak of the piriform activity (the first 50 ms inhale) and once using spikes counts from the entire inhalation.

To test the readout, each odor was presented at many concentrations (even though training was done for only one concentration). For the target odor, 100 trials were tested at each concentration (30 different concentrations ranging between 3% activated glomeruli to 30% activated glomeruli). Each trial that gave $\vec{w} \cdot \vec{r} > 0$ for the test odor was considered a correct classification. For each concentration, the percentage of trials that were correctly classified was calculated. Trials with non-target odors were tested as well, one trial for each odor at each concentration. All the non-target odors were correctly classified as not target ($\vec{w} \cdot \vec{r} < 0$) across all concentrations. The testing procedure was done using both the peak and full activity vectors, using the corresponding readout weight vectors.

## Concentration classification according to rate and latency of peak responses

We used the pyramidal cell peak rate responses to identify the concentration of bulb input. In each trial, pyramidal activity was characterized using two quantities, the rate of activity at the peak of response, $r_{peak}$, and the latency to the peak of the response from inhalation onset, $t_{peak}$. We recoded these two features for 500 trials of a target odor in 27 concentrations, spaced equally between 3% and 27% active glomeruli (500*27 trials in total). Because we are interested in understanding whether a concentration can be identified from peak properties for a specific odor, all trials used a single target odor. As explained above, since we generate odors randomly and all model mitral cells behave similarly, the results are independent of the choice of the target odor. For all of our classification methods, 250 trials at each concentration were used for training the classifier and the remaining 250 trials were used for testing. Because identifying the number of active glomeruli that drives the response depends on the differences between the percentages of active glomeruli (small differences are harder to detect) we chose to train and test responses within $\pm 3\%$ of active glomeruli relative to the target concentration. This is small enough (one tenth of the full studied range) to show identification of concentration from peak properties and large enough to allow for training and testing.

We considered a number of different classifications:

1. Classification based on peak rate, $r_{peak}$: For each target concentration we determined a value of $r_c$ that optimally separates our training set of lower concentrations, with $r_{peak} < r_c$, from those with higher concentration and $r_{peak} > r_c$. We then measured the percentage of trials from our testing set that were classified correctly using this value of $r_c$.
2. Classification based on peak latency, $t_{peak}$: The classification procedure was similar to (1), except that we determined $t_c$ (instead of $r_c$) to distinguish lower concentrations with $t_{peak} > t_c$ from higher concentration with $t_{peak} < t_c$.
3. Linear classification based on peak rate, $r_{peak}$, and peak latency, $t_{peak}$: Similar to (1), except we searched for two parameters, $a_c$ and $b_c$ (by searching exhaustively in the plane) such that the line $t = a_c r + b_c$ separated lower concentrations with $t_{peak} > a_c r_{peak} + b_c$ from higher concentration with $t_{peak} < a_c r_{peak} + b_c$.

4. Non-linear (log) classification based on peak rate, $r_{peak}$, and peak latency, $t_{peak}$: Similar to (3), except that we searched for a separating line of the form $t = a_c \log(r - b_c)$.

5. Clustering: For a pair of peak rates and latencies ($r_{peak}$, $t_{peak}$) from each test trial, we calculated all the (Euclidian) distances to pairs from all training trials. The concentration assigned to a test trial corresponded to the minimum average distance from training trials at that concentration. If the assigned concentration was within 4% of active glomeruli from the correct percentage of active glomeruli, the classification was considered correct. For each concentration, we calculated the percentage of test trials that were assigned correctly.

6. Classification based on spike counts, $s_{total}$: Classification was done as in (1) using the total number of spikes emitted by the full pyramidal population (independent of any peak property), with a value $s_c$ that separated lower concentrations with $s_{total} < s_c$ from higher concentrations with $s_{total} > s_c$.

## Experiments

All experimental protocols were approved by Duke University Institutional Animal Care and Use Committee. The methods for head-fixation, data acquisition, electrode placement, stimulus delivery, and analysis of single-unit and population odor responses are adapted from those described in detail previously (*Bolding and Franks, 2017*).

## Mice

Mice were adult (>P60, 20–24 g) offspring (four males, two females) of Emx1-cre (+/+) breeding pairs obtained from The Jackson Laboratory (005628). Mice were singly-housed on a normal light-dark cycle. Mice were habituated to head-fixation and tube restraint for 15–30 min on each of the two days prior to experiments. The head post was held in place by two clamps attached to ThorLabs posts. A hinged 50 ml Falcon tube on top of a heating pad (FHC) supported and restrained the body in the head-fixed apparatus.

## Data acquisition

Electrophysiological signals were acquired with a 32-site polytrode acute probe (A1 × 32-Poly3-5mm-25s-177, Neuronexus) through an A32-OM32 adaptor (Neuronexus) connected to a Cereplex digital headstage (Blackrock Microsystems). Unfiltered signals were digitized at 30 kHz at the headstage and recorded by a Cerebus multichannel data acquisition system (BlackRock Microsystems). Experimental events and respiration signal were acquired at 2 kHz by analog inputs of the Cerebus system. Respiration was monitored with a microbridge mass airflow sensor (Honeywell AWM3300V) positioned directly opposite the animal's nose. Negative airflow corresponds to inhalation and produces negative changes in the voltage of the sensor output.

## Electrode placement

For piriform cortex recordings, the recording probe was positioned in the anterior piriform cortex using a Patchstar Micromanipulator (Scientifica, UK), with the probe positioned at 1.32 mm anterior and 3.8 mm lateral from bregma. Recordings were targeted 3.5–4 mm ventral from the brain surface at this position with adjustment according to the local field potential (LFP) and spiking activity monitored online. Electrode sites on the polytrode span 275 µm along the dorsal-ventral axis. The probe was lowered until a band of intense spiking activity covering 30–40% of electrode sites near the correct ventral coordinate was observed, reflecting the densely packed layer II of piriform cortex. For simultaneous ipsilateral olfactory bulb recordings, a micromanipulator holding the recording probe was set to a 10-degree angle in the coronal plane, targeting the ventrolateral mitral cell layer. The probe was initially positioned above the center of the olfactory bulb (4.85 AP, 0.6 ML) and then lowered along this angle through the dorsal mitral cell and granule layers until a dense band of high-frequency activity was encountered, signifying the targeted mitral cell layer, typically between 1.5 and 2.5 mm from the bulb surface.

## Spike sorting and waveform characteristics

Individual units were isolated using Spyking-Circus (https://github.com/spyking-circus). Clusters with >1% of ISIs violating the refractory period (<2 ms) or appearing otherwise contaminated were manually removed from the dataset. Pairs of units with similar waveforms and coordinated refractory

periods in the cross-correlogram were combined into single clusters. Unit position with respect to electrode sites was characterized as the average of all electrode site positions weighted by the wave amplitude on each electrode.

## Acknowledgements

We thank Alexander Fleischmann and Andreas Schaefer for comments on the manuscript.

## Additional information

### Funding

| Funder | Grant reference number | Author |
|---|---|---|
| National Science Foundation | NeuroNex Award DBI-1707398 | LF Abbott |
| Simons Foundation | Collaboration on the Global Brain | LF Abbott |
| National Institute on Deafness and Other Communication Disorders | DC015525 | Kevin M Franks |
| National Institute on Deafness and Other Communication Disorders | DC016782 | Kevin M Franks |
| Gatsby Charitable Foundation | DBI-1707398 | LF Abbott |

The funders had no role in study design, data collection and interpretation, or the decision to submit the work for publication.

### Author contributions

Merav Stern, Conceptualization, Data curation, Software, Formal analysis, Validation, Investigation, Visualization, Methodology, Writing—original draft, Writing—review and editing; Kevin A Bolding, Data curation, Formal analysis, Investigation, Methodology, Writing—review and editing; LF Abbott, Kevin M Franks, Conceptualization, Resources, Data curation, Software, Formal analysis, Supervision, Funding acquisition, Validation, Investigation, Visualization, Methodology, Writing—original draft, Project administration, Writing—review and editing

### Author ORCIDs

Kevin A Bolding http://orcid.org/0000-0002-2271-5280
Kevin M Franks http://orcid.org/0000-0002-6386-9518

### Ethics

Animal experimentation: All experimental protocols were approved by Duke University Institutional Animal Care and Use Committee (protocol # A220-15-08), which was approved on 08-27-2015.

### Decision letter and Author response

Decision letter https://doi.org/10.7554/eLife.34831.016
Author response https://doi.org/10.7554/eLife.34831.017

## Additional files

### Supplementary files

• Source code 1. This is the code used to generate the model. This C code is used in an environment that can execute consecutive single steps and plot the results (e.g. xcode).
DOI: https://doi.org/10.7554/eLife.34831.012

• Source code 2. App Piriform model. This compiled program launches and runs the piriform model used here as an app. Parameters are described in the Materials and methods.
DOI: https://doi.org/10.7554/eLife.34831.013

• Transparent reporting form
DOI: https://doi.org/10.7554/eLife.34831.014

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
