## [Decision Letter]

[Editors’ note: a previous version of this study was rejected after peer review, but the authors submitted for reconsideration. The first decision letter after peer review is shown below.]

Thank you for submitting your work entitled "A transformation from temporal to ensemble coding in a model of piriform cortex" for consideration by *eLife*. Your article has been reviewed by three peer reviewers, and the evaluation has been overseen by a Reviewing Editor (Naoshige Uchida) and a Senior Editor. The reviewers have opted to remain anonymous.

Our decision has been reached after consultation between the reviewers. Based on these discussions and the individual reviews below, we regret to inform you that your work will not be considered further for publication in *eLife*.

Summary:

The authors explore the neural circuit mechanism underlying the transformation of odor information from the olfactory bulb (OB) to the piriform cortex (PCx) using a large scale spiking network model of the PCx. The model consists of a large number (>10,000) of integrate-and-fire neurons with current-based synaptic inputs, incorporating spike-latency based OB input, feedforward inhibitory neurons (FFI), feedback inhibitory neurons (FBI), and recurrent excitation. First, the authors show that the model can reproduce responses resembling those observed in experiments in terms of sparseness, temporal pattern, as well as concentration-dependent shift in peak timing and size. The authors show that PCx neurons responded with transient response followed by less active state. Because of this, a short latency OB input has larger impacts on PCx responses compared to a later OB input. By examining how each circuit element contributes to shape the dynamics of odor responses, the authors show that FBI and recurrent excitation play important roles in generating the transient PCx response dynamics.

The reviewers found that this manuscript addresses potentially important questions, and might provide interesting results. However and critically, they were unsure what the specific take home messages of the study are. They questioned: What can we really learn about the PCx from a seemingly generic circuit model that has omitted various known features of the PCx (e.g. see the reviewer 2's comments)? Is there anything that is not expected from the way that the model was designed? How robust the conclusions are to variations in the model parameters? The reviewers thought that the manuscript needs to be much improved with respect to these points.

Below we summarize the essential points reflecting also our discussion sessions in addition to the individual comments appended further below. Please note that individual comments (especially, those from reviewer 2) contain many important points to which we would like the authors to consider.

We are happy to look at a revised manuscript as far as the authors can fully address the essential points. Please note again, however, that the reviewers raised substantive concerns and it is critical that the authors address the essential points. In particular, please make sure to make a better use of modeling that allows the authors to draw well-constrained, novel conclusions.

Essential points:

1) Throughout the manuscript, the authors need to be more explicit about (i) what the exact observations are that need explanation, (ii) what the assumptions are they are making and (iii) what they find and conclude. Reviewer 2 mentioned that "most of the conclusions within Results paragraphs are of this nature – the sort of throwaway interpretive comments that often fill the tail ends of Discussion sections. The value of modeling is to quantify and/or challenge hypotheses that are not simple enough to be otherwise obvious. Modeling results deserve and require the same critical assessment as experimental results – do they really mean what they seem to, or are there extraneous variables that dominate the outcome? Are we predetermining the outcome by the way in which we set up the experiment, bypassing the actual question of interest?" The reviewers think that these points need to be addressed very clearly.

2) Related to the above issue, the reviewers are concerned whether the conclusions that the authors make are robust. Reviewer 1 mentioned that "While it seems that simple circuit features are enough to explain core features of the observed physiology (to be spelled out more explicitly, see above), the authors do not provide any evidence that indeed their findings are robust against variations in the (large number of) parameters they need to put into their model. One way to test this would be to somehow reduce their main findings to a few key output values (e.g. peak of firing rate with and without recurrent / FBI, late/early firing rate for FF only vs control, same/diff ratio as in Figure 5F and various other parameters extracted from their core findings) and assess, how robust these parameters are against variations in model parameter. Without such analysis it is difficult to assess how fundamental their findings are".

Reviewer 2 raises a similar concern, and provides some ways to address this issue with respect to normalization: "But, setting that aside, what do we learn from the present model? The authors favor FBI as the main normalizing force, which is fine. But this isn't really a finding. First, 'normalization' implies FBI, as you can't truly normalize without feedback. However, nominal FFI circuits often embed feedback effects, as do those in the present model in which broad MC->FFI projections and FFI-FFI interconnectivity are likely to produce some sort of global quasi-average activity level among FF interneurons for delivery onto PCx pyramidal cells that can serve as quasi-normalization so long as the gains are roughly matched. But more to the point, to conclude that FBI is the dominant effect requires exploration of the parameter spaces. Under what circumstances is this so? What if FBI were weaker, and FFI stronger? What if FFI were sigmoidal in effect and very strong, so that it reliably constrains PCx activation within a narrow range? How about some assessment of the innate requirement for both, because tuned FFI ensures that the pyramidal cells can respond within their dynamic ranges whereas FBI can further tune Pyr cell activity (surely for more interesting purposes that fine-tuning concentration normalization, though)". Overall, we would like the authors to demonstrate more explicitly throughout the manuscript that each conclusion is not a trivial consequence of the particular way the model was built (or the particular combination of parameters picked).

3) Reviewer 2 mentioned that "The interpretation (finding?) that FFI is subtractive is particularly baffling – of course it is so in the model, because the authors' LIF inhibitory synapses are subtractive. The GABA(A) synapses in the actual PCx will be predominantly divisive". Because the integrate-and-fire model used in this study uses subtractive voltage changes by synaptic inputs, it is likely that some of the subtractive nature of firing in the model is a direct consequence of this property of the model. The authors should make a strong case why this observation is interesting.

4) There was some disagreement among the reviewers as to what details need to be incorporated into modeling. These include oscillations at the β and γ frequencies, OB-PCx interactions and important properties or functions of the olfactory circuit. During discussion, we agreed that OB-PCx interaction is beyond the scope of the study. Nonetheless, there appear to be some ways to test the impact of oscillations such as entraining OB input with varying levels of oscillations. For other factors, we would like the authors to discuss how some of the omitted features may or may not affect their main conclusions.

5) While the authors show that the simple PCx circuitry is capable of reproducing many observed features they don't make explicit predictions. Figure 4 does contain implicit predictions about the role of the different network components but it would be very helpful and substantially strengthen the manuscript if these were made explicit in the Discussion. Direct experimental data to test the role of e.g. recurrent / feedback circuits or FFI would greatly strengthen the conclusions. Please pay particular attention to which modeling results may actually be attributable to piriform cortical function and which are simply epiphenomena of model parameters that are not genuinely constrained with respect to the biological circuit. The former require explanation, whereas the latter should be identified as such and not be reported as findings

6) Figure 4. The case without just recurrent excitation is lacking in the simulation (the case without just feedback inhibition is described in the text and this is fine). Because "no recurrent excitation" condition is explored in Figure 5, providing the result of simulation under this condition is important. Also, it seems inconsistent that the authors go back to examine the "no recurrent excitation and no feedback inhibition" condition in Figure 6. Related to this, the description in the text and the figure legend is not matching. The text says that the condition is "no recurrent excitation and no feedback inhibition" but the legend for Figure 6 says "no recurrent excitation". To display the results of both cases, it is clearer to use different colors for these two conditions.

7) Figure 6. Panel C shows the percent of responding cells and panel D shows total spikes generated by the entire population. These together seem to indicate that the spike count of individual pyramidal neurons does not change much with concentration, but this is unclear. It is useful to present the distribution of spikes in individual neurons. This is also necessary to support the notion: "…, indicating that population synchrony can provide a robust representation of odor concentration". To state that synchrony is enhanced, we need to know that the rate of individual neurons is not changed.

8) "Thus, the early peak in the PCx response can be used to rapidly decode both odor identity and concentration". The data supporting this conclusion is not supplied in the manuscript. The authors need to perform decoding using the peak rate. Also, although the text in the first paragraph of the subsection “Strategies for encoding odor intensity” says that the concentration-dependent change in latency to peak was modest, Figure 7B shows a substantial change and this is one of the major conclusions of the recent physiological study conducted by the authors (Bolding and Franks, *eLife* (2017)). Therefore, it is recommended to perform decoding using the latency to peak as well and compare the result with that using the peak rate.

The original review comments are appended below. It contains further elaborations of the issues summarized above. Please take these comments into account when you prepare a revision.

Reviewer #1:

This manuscript by Franks et al. presents a large-scale model of pcx incorporating OB input, FFI, FBI and recurrent excitation. It is very well written and aims to demonstrate that already the core features of PCX circuitry (as spelled out above) can explain key physiological observations, namely the reformatting of activity from OB to PCX as observed in in vivo physiology experiments. So, for me the main conclusion of the paper is that no "magic", i.e. unknown cell types or circuit motifs or complex temporal dynamics are needed to explain the reformatting from OB activity to PCX activity.

1) While spelled out quite nicely in the last paragraph of the Introduction and in the first paragraph of the Discussion, the authors need to be even more explicit about (i) what the exact observations are that need explanation, (ii) what the assumptions are they are making and (iii) what they find and conclude.

Most of this is there in the text but clarity and bullet points in Introduction and Discussion would help to gauge the quality and the far-reaching implications of their conclusions.

2) While it seems that simple circuit features are enough to explain core features of the observed physiology (to be spelled out more explicitly, see above), the authors do not provide any evidence that indeed their findings are robust against variations in the (large number of) parameters they need to put into their model. One way to test this would be to somehow reduce their main findings to a few key output values (e.g. peak of firing rate with and without recurrent / FBI, late/early firing rate for FF only vs control, same/diff ratio as in Figure 5F and various other parameters extracted from their core findings) and assess, how robust these parameters are against variations in model parameter. Without such analysis it is difficult to assess how fundamental their findings are.

3) While the authors show that the simple PCX circuitry is capable of reproducing many observed features they don't make explicit predictions. I guess Figure 4 does contain implicit predictions about the role of the different network components but it would be very helpful and substantially strengthen the manuscript if these were made explicit in the Discussion. Obviously, direct experimental data to test the role of e.g. recurrent / feedback circuits or FFI would massively strengthen the conclusions.

Reviewer #2:

I looked forward to reading this paper. Senior author Franks is an emerging force in the systems neurophysiology of piriform cortex who has earned respect from his colleagues, including myself, for his independent work. Joint senior author Abbott is a computational neuroscientist of the first rank. Respect to the first authors as well for their acceptance into these mentors' labs; I will look forward to their future work. But this manuscript is not something that any of these authors will want on their conscience. It is a loosely-vetted and ultimately generic set of simulations based on substantial apparent ignorance of the underlying systems and of the value and limitations of computational modeling as a research strategy. It is likely to have value as a starting point for computational exploration of the system, but none of the present findings offer insight; they are simple, deterministic outcomes of the LIF circuit motifs used, with all of the interesting and mysterious features of piriform cortex explicitly omitted from the study.

The concrete goals of the model are unclear, but the title indicates that the central finding is a transformation from temporal to ensemble coding, referring here to an emphasis on sequences of activity in OB generating characteristic 'spatial' activity patterns across PCx pyramidal neurons. To do this, the authors built a randomly connected network between LIF neurons labeled MCs and another labeled PCx pyramidal cells, plus two classes of PCx interneurons. The MC-PCx mapping determines an ensemble-to-ensemble transformation, with the 'temporal' aspect contributing by spreading out MC activity in time so that only the earliest activated MCs constitute the relevant 'ensemble'. The suppression of PCx pyramidal responses by FBIs contributes to this temporal selectivity, as later-activating MCs have a much reduced field of excitable PCx neurons to activate. Probably this effect would be even stronger if PCx pyramidal activity was scored within the first volley rather than over the full 200 ms stimulation (including after the rebound from FBI inhibition), but that's a detail. This is fine so far – a simple implementation of well-known response properties in PCx that could be a foundation for interesting further development. But most of what the authors proceed to at this point is just to activate this simple LIF circuit motif in straightforward ways and claim, implausibly, that they have learned something about the piriform cortex.

For example, there is the issue of timescale. The olfactory bulb is one of the canonical oscillogenic circuits of the brain, generating endogenous γ rhythmicity from unpatterned (or theta-patterned input arising from respiration and/or ET cell dynamics) input. OB-PCx interactions are coordinated in the β band, which supersedes γ via a transitional mechanism that remains unclear. These γ/β frequencies are generally the frequencies considered relevant for spike phase coding effects, as received and transformed by postsynaptic integration and plasticity mechanisms such as STDP. On the order of ten different labs over the years have dedicated substantial effort to measuring, analyzing, and understanding these fast OB (and to a lesser extent PCx) dynamics, both physiologically and in a coding sense. Many more groups pursue analogous neuronal systems dynamics questions in other structures, particularly hippocampus. Yet, in recent years, a corner of the olfaction literature has arisen that seems to not understand this body of work, or the relevance of the dynamical systems that underlie it, and has instead borrowed key concepts and catchphrases from this body of work and applied them naively to respiratory theta "oscillations". The deployment of citations in the introduction to this paper suggests that the authors are among those who do not understand this distinction, or appreciate its importance. For a dynamically-based model to simply omit mention of the central dynamics of this system is disqualifying. It renders the essential premise being tested in this paper incorrect and the outcome of the associated model all but irrelevant. This would be a critical problem even if the rest of the authors' approach was defensible.

This criticism may sound strange, because the Introduction is full of citations that, if skimmed, might appear to loosely support the premise constructed if one knew nothing of the literature omitted. But upon closer and more critical reading, the constructed premise does not stand. For example, some references note phenomenological responses as a side point to the chief goals of the corresponding paper. This manuscript uncritically interprets such reports as broadly vetted truths that will withstand the vagaries of sensory sampling in natural environments, even when a rudimentary understanding of physiological principles would make the error clear. Sometimes the reference in question bears responsibility for the error by overgeneralizing their findings or indulging a favorite hypothesis. Sometimes not. But the naïve rank-based temporal sequence hypotheses on the theta timescale for odor coding don't pass the sniff test. Response replicability under tightly controlled conditions is a deterministic near-tautology; it does not suffice to defend a hypothesis of coding. The entire central problem of olfaction is signal identification in the presence of copious and usually unpredictable noise; we know that virtually no easily-recognizable aspect of the signal is replicable in competing backgrounds, or in awake, behaving animals.

But if we ignore the timescale issue and accept the hypothesis of spike rank coding on a respiratory timescale (for sake of argument), along with the convergence and divergence of MC->Pyr connectivity, what will this offer us? The interesting features of PCx are its intrinsic dynamics, its emergent shared dynamics with OB, its hypothesized pattern-completion capabilities, and the idea that it may be a primary location for the integration of afferent odor input with existing odor memories. However, all of these are explicitly excluded from consideration by the present authors. This is problematic, because any one of these features, if implemented, would substantially dominate the core response properties of the piriform cortex to afferent input. Omitting all of them essentially predetermines that the outcome of the simulation will be irrelevant to any application. Constructing the model explicitly from LIF neurons further rules out the possibility of finding interesting biophysical mechanisms underlying PCx response properties.

But if we set these difficult problems aside, knowing that simulations also can serve as test beds for emergent circuit properties that may help us understand these more complex hypothesized functions, then what questions can the remaining model help us address? The authors answer (1) explaining spatially distributed ensemble codes for odor identity, (2) normalization across odor concentrations, and (3) the revelation of temporal codes for odor intensity.

Spatially distributed ensemble codes have been shown experimentally, of course. In the present model, the transform is essentially predetermined… different odors are defined by their different profiles of activation across glomeruli, and latencies of each. The divergence/convergence pattern onto PCx generate odor-specific patterns among PCx neurons: a generic spatial transform. This isn't a finding; it has been a foundational principle for decades. The selectivity for early MC responses is an idea, presumably motivated by rapid behavioral responses and fast functional adaptation in PCx pyramidal cells. I suppose hypothesizing that FBI feedback is primarily responsible for PCx pyramidal neuron response adaptation is potentially of interest, but this would need to be developed further.

"Thus, while our simplified PCx-like model does recapitulate the overall pattern of odor ensembles across PCx, ways in which it fails to do so makes specific predictions about cortical connectivity." This statement is fundamentally untrue. The model doesn't match the data, and the authors have some vague ideas about model limitations vis a vis reality that might bear on the difference. These aren't "specific predictions", especially because no effort at all is made to explore them. I'm not going to repeat myself more, but most of the conclusions within Results paragraphs are of this nature – the sort of throwaway interpretive comments that often fill the tail ends of Discussion sections. The value of modeling is to quantify and/or challenge hypotheses that are not simple enough to be otherwise obvious. Modeling results deserve and require the same critical assessment as experimental results – do they really mean what they seem to, or are there extraneous variables that dominate the outcome? Are we predetermining the outcome by the way in which we set up the experiment, bypassing the actual question of interest? This is really important to understand if one wants to be a consequential theoretical neuroscientist.

Normalization across odor concentrations. In sensory systems, it's just not reasonable to insert the main afferent intensity tolerance circuitry after the second synapse. Indeed, the authors acknowledge that most of this takes place earlier, substantially in the OB glomerular layer. But of course cortical circuits (almost?) all regulate local circuit input intensities, presumably because of the narrow dynamic ranges of neurons and the often narrower ranges of functional circuits. So it's not strictly wrong to refer to FFI and FBI in the PCx circuit as concentration normalization, but I would call it misleading. If you have negative feedback, then you have a capability that you could call 'intensity normalization' if you set up an experiment to show you that and had narrative reasons to call it that. But this is really different from a claim of "concentration normalization" in a chemosensory system.

But, setting that aside, what do we learn from the present model? The authors favor FBI as the main normalizing force, which is fine. But this isn't really a finding. First, 'normalization' implies FBI, as you can't truly normalize without feedback. However, nominal FFI circuits often embed feedback effects, as do those in the present model in which broad MC->FFI projections and FFI-FFI interconnectivity are likely to produce some sort of global quasi-average activity level among FF interneurons for delivery onto PCx pyramidal cells that can serve as quasi-normalization so long as the gains are roughly matched. But more to the point, to conclude that FBI is the dominant effect requires exploration of the parameter spaces. Under what circumstances is this so? What if FBI were weaker, and FFI stronger? What if FFI were sigmoidal in effect and very strong, so that it reliably constrains PCx activation within a narrow range? How about some assessment of the innate requirement for both, because tuned FFI ensures that the pyramidal cells can respond within their dynamic ranges whereas FBI can further tune Pyr cell activity (surely for more interesting purposes that fine-tuning concentration normalization, though). The interpretation (finding?) that FFI is subtractive is particularly baffling – of course it is so in the model, because the authors' LIF inhibitory synapses are subtractive. The GABA(A) synapses in the actual PCx will be predominantly divisive. It's a bug, not a feature.

The revelation of temporal codes for odor intensity I can't speak to. The effects emerge from a model that is missing most of the important features that would affect this finding, and that is not challenged by odor competition, correlated noise, or difficult discriminations. In the present model, this finding is an unavoidable consequence of the deterministic system and the particular models used for concentration (whether altering the number of glomeruli activated or stretching/compressing their activation in time). The model isn't constrained by olfactory physiology anywhere near well enough to generalize this effect to the real system.

Overall, this model is a generic spatial-to-spatial transformation circuit model with a temporal selectivity element added on. It is loosely based on the synaptology of piriform cortex (in the sense that it lumps several classes of neurons into larger groups) but dominated by standard FF and FB circuit motifs. Few surprises are possible because of the simple LIF models and straightforward connectivity maps. Diversity in cell types, biophysical properties, and the main interesting features of piriform cortex are omitted by design. So what is the point of the model? Just to reshape the odor representation again into a new form? We know that already, both from PCx data and from what randomly connected layers of LIF neurons do. What is the utility of this transformation?

The experiments associated with Figure 3 have the most potential; they start engaging with the dynamics of PCx circuitry. But there is a lot to do to actually assess the potential roles of these different interneuron classes, as opposed to generating a just-so story.

I want to be supportive of this work; I genuinely do. But to be supportive of the authors, which is ultimately more important, I instead offer this fairly harsh review. I'm sorry for that, but it is necessary.

*Reviewer #3:*

The aim of the manuscript was to understand the mechanisms underlying transformation of olfactory representations from the olfactory bulb (OB) to the piriform cortex (PCx) by creating a network model of PCx and manipulating each element of the network. First, the authors show that the model pyramidal neurons receiving odor-dependent variable-latency OB input can produce responses resembling those observed in experiments in terms of sparseness, temporal pattern, as well as concentration-dependent shift in peak timing and size. Then they simulate how each circuit element contributes to shape the dynamics of odor responses. Simulation indicated that the early OB input has a larger voice in generating the phasic excitatory response and this requires recurrent excitation and inhibition. Recurrent excitation was also shown to be central for more concentration-invariant odor representations in pyramidal neurons. This is a solid modeling study that can guide future efforts to better understand the circuit mechanisms of olfactory computations in the PCx. The level of the model seems to be adequate to address the specific aims and limitations of the model are also acknowledged. However, I have several issues that need to be addressed to fully support some of the authors' statements and to remove one inconsistency.

1) Figure 4. The case without just recurrent excitation is lacking in the simulation (the case without just feedback inhibition is described in the text and this is fine). Because "no recurrent excitation" condition is explored in Figure 5, providing the result of simulation under this condition is important. Also, it seems inconsistent that the authors go back to examine the "no recurrent excitation and no feedback inhibition" condition in Figure 6. Related to this, the description in the text and the figure legend is not matching. The text says that the condition is "no recurrent excitation and no feedback inhibition" but the legend for Figure 6 says "no recurrent excitation". To display the results of both cases, it is clearer to use different colors for these two conditions.

2) Figure 6. Panel C shows the percent of responding cells and panel D shows total spikes generated by the entire population. These together seem to indicate that the spike count of individual pyramidal neurons does not change much with concentration, but this is unclear. It is useful to present the distribution of spikes in individual neurons. This is also necessary to support the notion: "…, indicating that population synchrony can provide a robust representation of odor concentration". To state that synchrony is enhanced, we need to know that the rate of individual neurons is not changed.

3) "Thus, the early peak in the PCx response can be used to rapidly decode both odor identity and concentration". The data supporting this conclusion is not supplied in the manuscript. The authors need to perform decoding using the peak rate. Also, although the text in the first paragraph of the subsection “Strategies for encoding odor intensity” says that the concentration-dependent change in latency to peak was modest, Figure 7B shows a substantial change and this is one of the major conclusions of the recent physiological study conducted by the authors (Bolding and Franks, *eLife* (2017)). Therefore, it is recommended to perform decoding using the latency to peak as well and compare the result with that using the peak rate.

---

## [Author Response]

[Editors’ note: the author responses to the first round of peer review follow.]

Essential points:1) Throughout the manuscript, the authors need to be more explicit about (i) what the exact observations are that need explanation, (ii) what the assumptions are they are making and (iii) what they find and conclude. Reviewer 2 mentioned that "most of the conclusions within Results paragraphs are of this nature – the sort of throwaway interpretive comments that often fill the tail ends of Discussion sections. The value of modeling is to quantify and/or challenge hypotheses that are not simple enough to be otherwise obvious. Modeling results deserve and require the same critical assessment as experimental results – do they really mean what they seem to, or are there extraneous variables that dominate the outcome? Are we predetermining the outcome by the way in which we set up the experiment, bypassing the actual question of interest?" The reviewers think that these points need to be addressed very clearly.

We now provide experimental data (new Figure 1) that show how odor information is transformed from OB to PCx. Specifically, we show that while individual MTCs respond at different phases of the sniff cycle, most responsive neurons in PCx spike shortly after inhalation. This result indicates that early-responsive MTCs play a dominant role in driving PCx output, and that the impact of later-responding MTCs is suppressed in PCx. The goal of this study is to understand the neural circuit processes that implement this transformation.

We show that a simplified PCx circuit can do this with, as reviewer 1 says, ‘no magic’. In summary, we show that, and how, recurrent excitation and feedback inhibition play a major role in implementing this transformation. We have now made this goal more explicit up front, and we have been more clear about the assumptions we make to arrive at our conclusion and how these assumptions may influence the interpretation of our results.

We agree with reviewer 2 that modeling studies can be valuable for “quantifying and/or challenging hypotheses that are not simple enough to be otherwise obvious*”*. However, modeling studies can also be valuable in demonstrating the sufficiency of simple solutions to potentially complicated problems. Our study falls into this latter category.

2) Related to the above issue, the reviewers are concerned whether the conclusions that the authors make are robust. Reviewer 1 mentioned that "While it seems that simple circuit features are enough to explain core features of the observed physiology (to be spelled out more explicitly, see above), the authors do not provide any evidence that indeed their findings are robust against variations in the (large number of) parameters they need to put into their model. One way to test this would be to somehow reduce their main findings to a few key output values (e.g. peak of firing rate with and without recurrent / FBI, late/early firing rate for FF only vs control, same/diff ratio as in Figure 5F and various other parameters extracted from their core findings) and assess, how robust these parameters are against variations in model parameter. Without such analysis it is difficult to assess how fundamental their findings are".Reviewer 2 raises a similar concern, and provides some ways to address this issue with respect to normalization: "But, setting that aside, what do we learn from the present model? The authors favor FBI as the main normalizing force, which is fine. But this isn't really a finding. First, 'normalization' implies FBI, as you can't truly normalize without feedback. However, nominal FFI circuits often embed feedback effects, as do those in the present model in which broad MC->FFI projections and FFI-FFI interconnectivity are likely to produce some sort of global quasi-average activity level among FF interneurons for delivery onto PCx pyramidal cells that can serve as quasi-normalization so long as the gains are roughly matched. But more to the point, to conclude that FBI is the dominant effect requires exploration of the parameter spaces. Under what circumstances is this so? What if FBI were weaker, and FFI stronger? What if FFI were sigmoidal in effect and very strong, so that it reliably constrains PCx activation within a narrow range? How about some assessment of the innate requirement for both, because tuned FFI ensures that the pyramidal cells can respond within their dynamic ranges whereas FBI can further tune Pyr cell activity (surely for more interesting purposes that fine-tuning concentration normalization, though)". Overall, we would like the authors to demonstrate more explicitly throughout the manuscript that each conclusion is not a trivial consequence of the particular way the model was built (or the particular combination of parameters picked).

Following reviewer 1’s suggestion, we used population firing rates to examine the robustness while varying vales of many of the model’s key parameters. These analyses demonstrate that our model is robust over a reasonable range of parameter values, as well as identify the parameters to which the model is most sensitive. These analyses have also yielded some additional novel and interesting results which we elaborate in the text.

As one example, we originally said that runaway excitation occurs when FBI is removed and recurrent excitation is left in place. However, FBI is disynaptic, with an excitatory connection from pyramidal cells onto FBI neurons and an inhibitory connection from FBI neurons back onto pyramidal cells. Completely eliminating either indeed produces runaway excitation. However, changing these two variables has quite distinct effects on the population response. Gradually varying the strength of connections from pyramidal cells into FBI neurons produces a corresponding gradual change in the pyramidal cell spiking, as we expected. However, pyramidal cell population spiking is largely robust to changes in FBI. This occurs because a small increase in FBI strength silences more FBI neurons (since they are also interconnected) resulting in fewer active FBI neurons, which stabilizes the overall amount of output inhibition into the pyramidal cells. However, we did notice that as FBI gets stronger, pyramidal cell output become synchronized at ~20 Hz. We discuss these results further in response to Essential point no. 4.

3) Reviewer 2 mentioned that "The interpretation (finding?) that FFI is subtractive is particularly baffling – of course it is so in the model, because the authors' LIF inhibitory synapses are subtractive. The GABA(A) synapses in the actual PCx will be predominantly divisive". Because the integrate-and-fire model used in this study uses subtractive voltage changes by synaptic inputs, it is likely that some of the subtractive nature of firing in the model is a direct consequence of this property of the model. The authors should make a strong case why this observation is interesting.

Whether and why different types of GABAergic inhibition have subtractive or divisive effects is currently an area of intense interest, including in PCx. Differences in these types of operations are thought to depend on the types of inhibitory interneurons (e.g. SOM vs. PV cells) and their target sites on the postsynaptic cell (i.e. dendrite- vs. soma-targeting). For example, Sturgill and Isaacson (2015) recently showed that SOM-mediated inhibition in PCx is almost completely subtractive while PV-mediated inhibition is largely divisive inhibition.

Our model has two types of inhibition, FFI and FBI, that are implemented the same way, as subtractive voltage changes in LIF neurons. However, these play very different roles in transforming OB input. We showed that the slope of the population input-output relationship (i.e. gain) is steeper when recurrent excitation/FBI is removed (leaving only FFI), indicating that recurrent excitation/FBI effectively controls gain while FFI is relatively ineffective at doing so (Figure 8). By contrast, gain barely changes when FFI is removed (leaving recurrent excitation/FBI), indicating that FFI’s contribution is predominantly subtractive. We have now also added an extensive series of simulations in which we vary the strengths of the different parameters controlling either FFI or FBI (Figures 5, 6). These all support our conclusion general that FFI is relatively ineffective and largely subtractive, and that output depends much more steeply on FBI.

4) There was some disagreement among the reviewers as to what details need to be incorporated into modeling. These include oscillations at the β and γ frequencies, OB-PCx interactions and important properties or functions of the olfactory circuit. During discussion, we agreed that OB-PCx interaction is beyond the scope of the study. Nonetheless, there appear to be some ways to test the impact of oscillations such as entraining OB input with varying levels of oscillations. For other factors, we would like the authors to discuss how some of the omitted features may or may not affect their main conclusions.

We have now added two sets of simulations. In one, we simply show that, when using our default parameters, entraining OB input with varying strengths of β oscillations (20 Hz) drives corresponding oscillatory output in PCx (Figure 5—figure supplement 2). However, in our parameter validation, we found that PCx output becomes strong oscillatory at ~20 Hz when feedback inhibition is very strong (Figure 5). We therefore also asked whether these oscillations persist when OB input is also oscillatory. We show that, with moderate feedback inhibition, PCx spiking follows oscillating OB input over a range of frequencies (20-80 Hz). However, with strong feedback inhibition, PCx output always oscillates around 20 Hz (Figure 5—figure supplement 3). We thought this result was interesting because, as reviewer 2 points out, odor responses in PCx often evoke β activity, although the origins of this activity remain poorly understood.

These results therefore provide some insight into the origin of odor-evoked β oscillations in PCx. We also note that centrifugal OB-to-PCx projections could drive 20 Hz oscillations in OB.

*5) While the authors show that the simple PCx circuitry is capable of reproducing many observed features they don't make explicit predictions. Figure 4 does contain implicit predictions about the role of the different network components but it would be very helpful and substantially strengthen the manuscript if these were made explicit in the Discussion. Direct experimental data to test the role of e.g. recurrent / feedback circuits or FFI would greatly strengthen the conclusions. Please pay particular attention to which modeling results may actually be attributable to piriform cortical function and which are simply epiphenomena of model parameters that are not genuinely constrained with respect to the biological circuit. The former require explanation, whereas the latter should be identified as such and not be reported as findings.*

We have added a section to the Discussion that explicitly states some of the predictions that we can make from our model. Some confusion was generated by a part of the Results section that contextualized some results. Following reviewer 1’s suggestion, we have moved this to the Discussion section. This section was also a source of confusion for reviewer 2 (please see below). We have therefore been more clear about which of these results follow necessarily from the way we constructed the model and which make testable predictions about the anatomical and functional organization of PCx. We are in the process of performing some of the experiments required to test the specific roles of recurrent and FBI circuits, and our preliminary results are very encouraging. However, these data are beyond the scope of this manuscript.

6) Figure 4. The case without just recurrent excitation is lacking in the simulation (the case without just feedback inhibition is described in the text and this is fine). Because "no recurrent excitation" condition is explored in Figure 5, providing the result of simulation under this condition is important. Also, it seems inconsistent that the authors go back to examine the "no recurrent excitation and no feedback inhibition" condition in Figure 6. Related to this, the description in the text and the figure legend is not matching. The text says that the condition is "no recurrent excitation and no feedback inhibition" but the legend for Figure 6 says "no recurrent excitation". To display the results of both cases, it is clearer to use different colors for these two conditions.

When we labeled “no rec” we meant ‘no recurrent excitation *and* no feedback inhibition’; that is, we were not considering the case in which only recurrent excitation was removed. We apologize for this misunderstanding and we have corrected the labeling. Note, however, that we now also do examine the “no recurrent excitation only” condition when we vary recurrent excitation onto pyramidal cells only (Figure 6), as well as no FBI only when we vary recurrent excitation onto FBI neurons only and FBI itself.

7) Figure 6. Panel C shows the percent of responding cells and panel D shows total spikes generated by the entire population. These together seem to indicate that the spike count of individual pyramidal neurons does not change much with concentration, but this is unclear. It is useful to present the distribution of spikes in individual neurons. This is also necessary to support the notion: "…, indicating that population synchrony can provide a robust representation of odor concentration". To state that synchrony is enhanced, we need to know that the rate of individual neurons is not changed.

The reviewer is quite correct and this is an important point that we need to make clearly to support our conclusions. We have now provided an additional panel to this figure (now Figure 7F) that shows the distribution of spikes per neuron at multiple concentrations. Indeed, we show that the number of spikes a responsive neuron fires does not change substantively with concentration.

8) "Thus, the early peak in the PCx response can be used to rapidly decode both odor identity and concentration". The data supporting this conclusion is not supplied in the manuscript. The authors need to perform decoding using the peak rate. Also, although the text in the first paragraph of the subsection “Strategies for encoding odor intensity” says that the concentration-dependent change in latency to peak was modest, Figure 7B shows a substantial change and this is one of the major conclusions of the recent physiological study conducted by the authors (Bolding and Franks, eLife (2017)). Therefore, it is recommended to perform decoding using the latency to peak as well and compare the result with that using the peak rate.

This is a great suggestion. We have added concentration classification using population spike count, peak rate, and latency to peak, as well as combinations of both peak rate and latency to peak (Figure 9F).

The original review comments are appended below. It contains further elaborations of the issues summarized above. Please take these comments into account when you prepare a revision.Reviewer #1:[…] 1) While spelled out quite nicely in the last paragraph of the Introduction and in the first paragraph of the Discussion, the authors need to be even more explicit about (i) what the exact observations are that need explanation, (ii) what the assumptions are they are making and (iii) what they find and conclude.Most of this is there in the text but clarity and bullet points in Introduction and Discussion would help to gauge the quality and the far-reaching implications of their conclusions.

We thank the reviewer for making this point clearly. We have opted not to include bullet points in the Introduction and Discussion, but we have made these points much more clearly. We have also added new experimental data showing the OB-PCx transformation directly (Figure 1), and we explicitly state that the goal of this study is to understand how this transformation in implemented.

2) While it seems that simple circuit features are enough to explain core features of the observed physiology (to be spelled out more explicitly, see above), the authors do not provide any evidence that indeed their findings are robust against variations in the (large number of) parameters they need to put into their model. One way to test this would be to somehow reduce their main findings to a few key output values (e.g. peak of firing rate with and without recurrent / FBI, late/early firing rate for FF only vs control, same/diff ratio as in Figure 5F and various other parameters extracted from their core findings) and assess, how robust these parameters are against variations in model parameter. Without such analysis it is difficult to assess how fundamental their findings are.

We have now provided an extensive set of analyses that examine model performance (e.g. population firing rate) over a range of values for multiple parameters and sets of parameters. Briefly, we show that FFI provides a generalized suppression of model performance but doesn’t “shape” the PCx response. By contrast, the PCx output is very sensitive to a balance of recurrent excitation and FBI. In fact, because of these analyses, we made an interesting observation: We now show how changing the recruitment of FBI neurons or the FBI itself have very distinct impacts of population activity, and we thank the reviewer for these helpful suggestions.

3) While the authors show that the simple PCX circuitry is capable of reproducing many observed features they don't make explicit predictions. I guess Figure 4 does contain implicit predictions about the role of the different network components but it would be very helpful and substantially strengthen the manuscript if these were made explicit in the Discussion. Obviously, direct experimental data to test the role of e.g. recurrent / feedback circuits or FFI would massively strengthen the conclusions.

We have now added a section to the Discussion that makes explicit the testable predictions made by our model.

Reviewer #2:I looked forward to reading this paper. Senior author Franks is an emerging force in the systems neurophysiology of piriform cortex who has earned respect from his colleagues, including myself, for his independent work. Joint senior author Abbott is a computational neuroscientist of the first rank. Respect to the first authors as well for their acceptance into these mentors' labs; I will look forward to their future work. But this manuscript is not something that any of these authors will want on their conscience. It is a loosely-vetted and ultimately generic set of simulations based on substantial apparent ignorance of the underlying systems and of the value and limitations of computational modeling as a research strategy. It is likely to have value as a starting point for computational exploration of the system, but none of the present findings offer insight; they are simple, deterministic outcomes of the LIF circuit motifs used, with all of the interesting and mysterious features of piriform cortex explicitly omitted from the study.The concrete goals of the model are unclear, but the title indicates that the central finding is a transformation from temporal to ensemble coding, referring here to an emphasis on sequences of activity in OB generating characteristic 'spatial' activity patterns across PCx pyramidal neurons. To do this, the authors built a randomly connected network between LIF neurons labeled MCs and another labeled PCx pyramidal cells, plus two classes of PCx interneurons. The MC-PCx mapping determines an ensemble-to-ensemble transformation, with the 'temporal' aspect contributing by spreading out MC activity in time so that only the earliest activated MCs constitute the relevant 'ensemble'. The suppression of PCx pyramidal responses by FBIs contributes to this temporal selectivity, as later-activating MCs have a much reduced field of excitable PCx neurons to activate. Probably this effect would be even stronger if PCx pyramidal activity was scored within the first volley rather than over the full 200 ms stimulation (including after the rebound from FBI inhibition), but that's a detail. This is fine so far – a simple implementation of well-known response properties in PCx that could be a foundation for interesting further development. But most of what the authors proceed to at this point is just to activate this simple LIF circuit motif in straightforward ways and claim, implausibly, that they have learned something about the piriform cortex.For example, there is the issue of timescale. The olfactory bulb is one of the canonical oscillogenic circuits of the brain, generating endogenous γ rhythmicity from unpatterned (or theta-patterned input arising from respiration and/or ET cell dynamics) input. OB-PCx interactions are coordinated in the β band, which supersedes γ via a transitional mechanism that remains unclear. These γ/β frequencies are generally the frequencies considered relevant for spike phase coding effects, as received and transformed by postsynaptic integration and plasticity mechanisms such as STDP. On the order of ten different labs over the years have dedicated substantial effort to measuring, analyzing, and understanding these fast OB (and to a lesser extent PCx) dynamics, both physiologically and in a coding sense. Many more groups pursue analogous neuronal systems dynamics questions in other structures, particularly hippocampus. Yet, in recent years, a corner of the olfaction literature has arisen that seems to not understand this body of work, or the relevance of the dynamical systems that underlie it, and has instead borrowed key concepts and catchphrases from this body of work and applied them naively to respiratory theta "oscillations". The deployment of citations in the introduction to this paper suggests that the authors are among those who do not understand this distinction, or appreciate its importance. For a dynamically-based model to simply omit mention of the central dynamics of this system is disqualifying. It renders the essential premise being tested in this paper incorrect and the outcome of the associated model all but irrelevant. This would be a critical problem even if the rest of the authors' approach was defensible.

Clearly, we do not share the reviewer’s perspective. Our goal here, which we now state much more clearly, is to understand the transformation odor information between OB and PCx that occurs within the first sniff after odor onset. Our relatively simple model circuit is able to implement this transformation, and by doing so, we make a number of testable predictions about the distinct roles that different elements the cortical circuit play.

Our work does not validate, invalidate, or speak to role that oscillations play in coding olfactory or other information; it is a study of a neural circuit operation at a different timescale. Second, by approximating experimental measures of mitral cell spiking, we are implicitly incorporating some of the bulb’s inherent activity patterns in our model. Of course, we acknowledge that we are also missing some of it too, and we discuss this. However, we have now explicitly examined the effect of entraining bulb output at β frequencies, and we now show how this alters the early (i.e. first sniff) cortical response. Finally, in the parameter analyses that we have now added, we show that we can induce short, odor-evoked β-frequency oscillations in piriform by strengthening the feedback inhibitory synapse. Taken together, we feel that this work constitutes a valuable contribution, and that our revisions addresses the major issues the reviewer raises here.

This criticism may sound strange, because the Introduction is full of citations that, if skimmed, might appear to loosely support the premise constructed if one knew nothing of the literature omitted. But upon closer and more critical reading, the constructed premise does not stand. For example, some references note phenomenological responses as a side point to the chief goals of the corresponding paper. This manuscript uncritically interprets such reports as broadly vetted truths that will withstand the vagaries of sensory sampling in natural environments, even when a rudimentary understanding of physiological principles would make the error clear. Sometimes the reference in question bears responsibility for the error by overgeneralizing their findings or indulging a favorite hypothesis. Sometimes not. But the naïve rank-based temporal sequence hypotheses on the theta timescale for odor coding don't pass the sniff test. Response replicability under tightly controlled conditions is a deterministic near-tautology; it does not suffice to defend a hypothesis of coding. The entire central problem of olfaction is signal identification in the presence of copious and usually unpredictable noise; we know that virtually no easily-recognizable aspect of the signal is replicable in competing backgrounds, or in awake, behaving animals.

It is difficult to respond to this point without reference to the specific citations that the reviewer thinks we are misinterpreting. We also do not agree with the reviewer’s contention that nothing can be learned about neural circuit function by studying, or simulating, sensory responses under tightly controlled conditions.

But if we ignore the timescale issue and accept the hypothesis of spike rank coding on a respiratory timescale (for sake of argument), along with the convergence and divergence of MC->Pyr connectivity, what will this offer us? The interesting features of PCx are its intrinsic dynamics, its emergent shared dynamics with OB, its hypothesized pattern-completion capabilities, and the idea that it may be a primary location for the integration of afferent odor input with existing odor memories. However, all of these are explicitly excluded from consideration by the present authors. This is problematic, because any one of these features, if implemented, would substantially dominate the core response properties of the piriform cortex to afferent input. Omitting all of them essentially predetermines that the outcome of the simulation will be irrelevant to any application. Constructing the model explicitly from LIF neurons further rules out the possibility of finding interesting biophysical mechanisms underlying PCx response properties.

We feel that the integration and transformation of elemental odor information from olfactory bulb (i.e. a combinatorial glomerular code) into a synthetic representation of the odor that is robust to some features of the stimulus, such as concentration, is also an interesting feature of PCx. The processes we study in this model not only do that, but will also provide a platform for future studies, which can include other features of PCx that reviewer 2 thinks are more interesting.

But if we set these difficult problems aside, knowing that simulations also can serve as test beds for emergent circuit properties that may help us understand these more complex hypothesized functions, then what questions can the remaining model help us address? The authors answer (1) explaining spatially distributed ensemble codes for odor identity, (2) normalization across odor concentrations, and (3) the revelation of temporal codes for odor intensity.Spatially distributed ensemble codes have been shown experimentally, of course. In the present model, the transform is essentially predetermined.… different odors are defined by their different profiles of activation across glomeruli, and latencies of each. The divergence/convergence pattern onto PCx generate odor-specific patterns among PCx neurons: a generic spatial transform. This isn't a finding; it has been a foundational principle for decades. The selectivity for early MC responses is an idea, presumably motivated by rapid behavioral responses and fast functional adaptation in PCx pyramidal cells. I suppose hypothesizing that FBI feedback is primarily responsible for PCx pyramidal neuron response adaptation is potentially of interest, but this would need to be developed further.

Here, again, we fear our failure to explicate the goals of this study has led to some confusion. Yes, the “spatial” aspect of our model’s transformation predetermined by the way we constructed the model, and we don’t claim that this is a finding. However, we do find (i) that neurons participating in these ensembles tend to be activated early even though OB input arrives throughout the sniff, and (ii) that different odors activate ~10% of PCx neurons across a range of input strengths. These findings are in accord with recent experimental observations (e.g. Stettler and Axel, 2009; Miura et al., 2012; Bolding and Franks 2017; Roland et al., 2017). The goal of the present study is to understand the circuit processes that underlie these observations. As above, in the revised manuscript we have now stated the goals of our study much more clearly, more clearly delineating between what are novel results of our simulations and what are the epiphenomena that follow necessarily from how we constructed the model.

"Thus, while our simplified PCx-like model does recapitulate the overall pattern of odor ensembles across PCx, ways in which it fails to do so makes specific predictions about cortical connectivity." This statement is fundamentally untrue. The model doesn't match the data, and the authors have some vague ideas about model limitations vis a vis reality that might bear on the difference. These aren't "specific predictions", especially because no effort at all is made to explore them. I'm not going to repeat myself more, but most of the conclusions within Results paragraphs are of this nature – the sort of throwaway interpretive comments that often fill the tail ends of Discussion sections. The value of modeling is to quantify and/or challenge hypotheses that are not simple enough to be otherwise obvious. Modeling results deserve and require the same critical assessment as experimental results – do they really mean what they seem to, or are there extraneous variables that dominate the outcome? Are we predetermining the outcome by the way in which we set up the experiment, bypassing the actual question of interest? This is really important to understand if one wants to be a consequential theoretical neuroscientist.

Here we think the reviewer is addressing the point about cross-odor correlations. In our model, a given odor activates about 10% of neurons distributed across the network, which does recapitulate experimental observations. However, we constructed our model assuming random connectivity between OB and PCx, which was justified by both course mapping experiments (e.g. Sosulski et al., 2011; Miyamichi et al., 2011; Ghosh et al., 2011; and others). We also assumed, as a starting point, random collateral connectivity. This produces random patterns of activity, which is consistent with experimental data that show no gross spatial topography (Stettler and Axel 2009; Miura et al., 2012; Roland et al., 2017). Consequently, cross-odor responses in the model have near-zero correlations; a “finding” that follows immediately from our starting assumptions. However, recent experimental studies have shown that cross-odor responses are relatively highly correlated (Otazu et al., 2015; Bolding and Franks 2017; Roland et al., 2017). This mismatch suggests that our starting assumptions of random connectivity – from OB to PCx, within PCx, or both – are likely incorrect. It would be great if we ‘solved’ this problem here, but that this is beyond the scope of this project. Instead, we describe our observation and lay out the problem and its implications. Following reviewer 1’s recommendation, we have moved this text to the Discussion.

Normalization across odor concentrations. In sensory systems, it's just not reasonable to insert the main afferent intensity tolerance circuitry after the second synapse. Indeed, the authors acknowledge that most of this takes place earlier, substantially in the OB glomerular layer. But of course cortical circuits (almost?) all regulate local circuit input intensities, presumably because of the narrow dynamic ranges of neurons and the often narrower ranges of functional circuits. So it's not strictly wrong to refer to FFI and FBI in the PCx circuit as concentration normalization, but I would call it misleading. If you have negative feedback, then you have a capability that you could call 'intensity normalization' if you set up an experiment to show you that and had narrative reasons to call it that. But this is really different from a claim of "concentration normalization" in a chemosensory system.But, setting that aside, what do we learn from the present model? The authors favor FBI as the main normalizing force, which is fine. But this isn't really a finding. First, 'normalization' implies FBI, as you can't truly normalize without feedback. However, nominal FFI circuits often embed feedback effects, as do those in the present model in which broad MC->FFI projections and FFI-FFI interconnectivity are likely to produce some sort of global quasi-average activity level among FF interneurons for delivery onto PCx pyramidal cells that can serve as quasi-normalization so long as the gains are roughly matched. But more to the point, to conclude that FBI is the dominant effect requires exploration of the parameter spaces. Under what circumstances is this so? What if FBI were weaker, and FFI stronger? What if FFI were sigmoidal in effect and very strong, so that it reliably constrains PCx activation within a narrow range? How about some assessment of the innate requirement for both, because tuned FFI ensures that the pyramidal cells can respond within their dynamic ranges whereas FBI can further tune Pyr cell activity (surely for more interesting purposes that fine-tuning concentration normalization, though). The interpretation (finding?) that FFI is subtractive is particularly baffling – of course it is so in the model, because the authors' LIF inhibitory synapses are subtractive. The GABA(A) synapses in the actual PCx will be predominantly divisive. It's a bug, not a feature.

We have performed an extensive series of additional simulations in which we vary many of the model’s key parameters. The specific ways in which circuit output changes are insightful, and some of these are now included in the revised manuscript. However, in reference to this comment, these simulations clearly indicate that our finding that FBI is the dominant factor responsible for control gain in PCx is robust across a wide range and combination of parameter values.

Also, it should be pointed out that there is no evidence that either FFI or FBI in PCx are tuned to specific odors. In fact, several studies have demonstrated quite conclusively that while odor responses in pyramidal cells are typically quite specific, both FFI and FBI are very broadly tuned (Poo and Isaacson, 2009; Hu et al., 2016; Bolding and Franks, 2017).

The revelation of temporal codes for odor intensity I can't speak to. The effects emerge from a model that is missing most of the important features that would affect this finding, and that is not challenged by odor competition, correlated noise, or difficult discriminations. In the present model, this finding is an unavoidable consequence of the deterministic system and the particular models used for concentration (whether altering the number of glomeruli activated or stretching/compressing their activation in time). The model isn't constrained by olfactory physiology anywhere near well enough to generalize this effect to the real system.

We respectfully disagree with reviewer 2’s perspective. Bolding and Franks (2017) showed that neither the number of responsive neurons nor the number of evoked spikes (per cell or across the population) increases with odor concentration, but that response latency decreases and population synchrony increases with increasing odorant concentrations. Here, we show that a simplified model is able to recapitulate this general result. Of course the actual system will be more complicated. But, to us, this indicates that many of the parameters that reviewer 2 claims our model requires may not be necessary to account for these observations.

Overall, this model is a generic spatial-to-spatial transformation circuit model with a temporal selectivity element added on. It is loosely based on the synaptology of piriform cortex (in the sense that it lumps several classes of neurons into larger groups) but dominated by standard FF and FB circuit motifs. Few surprises are possible because of the simple LIF models and straightforward connectivity maps. Diversity in cell types, biophysical properties, and the main interesting features of piriform cortex are omitted by design. So what is the point of the model? Just to reshape the odor representation again into a new form? We know that already, both from PCx data and from what randomly connected layers of LIF neurons do. What is the utility of this transformation?The experiments associated with Figure 3 have the most potential; they start engaging with the dynamics of PCx circuitry. But there is a lot to do to actually assess the potential roles of these different interneuron classes, as opposed to generating a just-so story.

We have now added an extensive series of parameter characterizations that address exactly this point.

I want to be supportive of this work; I genuinely do. But to be supportive of the authors, which is ultimately more important, I instead offer this fairly harsh review. I'm sorry for that, but it is necessary.

We thank the reviewer for this careful, detailed and critical analysis of our work. We have worked hard to address their points of concern and contention. We feel that we have addressed many of these, and that our manuscript is much improved because of this. Given our very different perspectives, we don’t expect a wholesale and enthusiastic endorsement of our approach, but we do hope the reviewer will agree that our manuscript is much improved and will be a valuable addition to the literature.

Reviewer #3:[…] However, I have several issues that need to be addressed to fully support some of the authors' statements and to remove one inconsistency.1) Figure 4. The case without just recurrent excitation is lacking in the simulation (the case without just feedback inhibition is described in the text and this is fine). Because "no recurrent excitation" condition is explored in Figure 5, providing the result of simulation under this condition is important. Also, it seems inconsistent that the authors go back to examine the "no recurrent excitation and no feedback inhibition" condition in Figure 6. Related to this, the description in the text and the figure legend is not matching. The text says that the condition is "no recurrent excitation and no feedback inhibition" but the legend for Figure 6 says "no recurrent excitation". To display the results of both cases, it is clearer to use different colors for these two conditions.

Unfortunately, a lot of confusion resulted from a simple mistake. As we said above, when we labeled “no rec” we meant ‘no recurrent excitation and no feedback inhibition’; that is, we were not considering the case in which only recurrent excitation was removed. We apologize for this misunderstanding and we have corrected the labeling. However, please note that in the revised manuscript we have included simulations in which we vary the strengths of individual parameters, including the strength of pyramidal recurrent excitation onto pyramidal cells only (Figure 6B).

2) Figure 6. Panel C shows the percent of responding cells and panel D shows total spikes generated by the entire population. These together seem to indicate that the spike count of individual pyramidal neurons does not change much with concentration, but this is unclear. It is useful to present the distribution of spikes in individual neurons. This is also necessary to support the notion: "…, indicating that population synchrony can provide a robust representation of odor concentration". To state that synchrony is enhanced, we need to know that the rate of individual neurons is not changed.

Absolutely! This panel was originally included, which is how we could make the case for enhanced synchrony, but it was then omitted in one of the subsequent revisions. Apologies, it’s back.

3) "Thus, the early peak in the PCx response can be used to rapidly decode both odor identity and concentration". The data supporting this conclusion is not supplied in the manuscript. The authors need to perform decoding using the peak rate. Also, although the text in the first paragraph of the subsection “Strategies for encoding odor intensity” says that the concentration-dependent change in latency to peak was modest, Figure 7B shows a substantial change and this is one of the major conclusions of the recent physiological study conducted by the authors (Bolding and Franks, eLife (2017)). Therefore, it is recommended to perform decoding using the latency to peak as well and compare the result with that using the peak rate.

As described above (Essential point #8), we have added these decoding analyses.